# State Sequences Prediction via Fourier Transform for Representation Learning

**Mingxuan Ye**[1] **Yufei Kuang**[1] **Jie Wang**[1,2*] **Rui Yang**[1]
**Wengang Zhou**[1,2] **Houqiang Li**[1,2] **Feng Wu**[1]

[1]CAS Key Laboratory of Technology in GIPAS, University of Science and Technology of China
[2]Institute of Artificial Intelligence, Hefei Comprehensive National Science Center
{mingxuanye, yfkuang, yr0013}@mail.ustc.edu.cn
{jiewangx, zhwg, lihq, fengwu}@ustc.edu.cn

## Abstract

While deep reinforcement learning (RL) has been demonstrated effective in solving complex control tasks, sample efficiency remains a key challenge due to the large amounts of data required for remarkable performance. Existing research explores the application of representation learning for data-efficient RL, e.g., learning predictive representations by predicting long-term future states. However, many existing methods do not fully exploit the structural information inherent in sequential state signals, which can potentially improve the quality of long-term decision-making but is difficult to discern in the time domain. To tackle this problem, we propose **S**tate Sequences **P**rediction via **F**ourier Transform (SPF), a novel method that exploits the frequency domain of state sequences to extract the underlying patterns in time series data for learning expressive representations efficiently. Specifically, we theoretically analyze the existence of structural information in state sequences, which is closely related to policy performance and signal regularity, and then propose to predict the Fourier transform of infinite-step future state sequences to extract such information. One of the appealing features of SPF is that it is simple to implement while not requiring storage of infinite-step future states as prediction targets. Experiments demonstrate that the proposed method outperforms several state-of-the-art algorithms in terms of both sample efficiency and performance.[2]

## 1 Introduction

Deep reinforcement learning (RL) has achieved remarkable success in complex sequential decision-making tasks, such as computer games [1], robotic control [2], and combinatorial optimization [3]. However, these methods typically require large amounts of training data to learn good control, which limits the applicability of RL algorithms to real-world problems. The crucial challenge is to improve the sample efficiency of RL methods. To address this challenge, previous research has focused on representation learning to extract adequate and valuable information from raw sensory data and train RL agents in the learned representation space, which has been shown to be significantly more data-efficient [4–8]. Many of these algorithms rely on auxiliary self-supervision tasks, such as predicting future reward signals [4] and reconstructing future observations [5], to incorporate prior knowledge about the environment into the representations.

*Corresponding Author

[2]The code of SPF is available on GitHub at https://github.com/MIRALab-USTC/RL-SPF/.

37th Conference on Neural Information Processing Systems (NeurIPS 2023).

Due to the sequential nature of RL tasks, multi-step future signals inherently contain more features that are valuable for long-term decision-making than immediate future signals. Recent work has demonstrated that leveraging future reward sequences as supervisory signals is effective in improving the generalization performance of visual RL algorithms [9, 10]. However, we argue that the state sequence provides a more informative supervisory signal compared to the sparse reward signal. As shown in the top portion of Figure 1, the sequence of future states essentially determines future actions and further influences the sequence of future rewards. Therefore, state sequences maximally preserve the influence of the transition intrinsic to the environment and the effect of actions generated from the current policy.

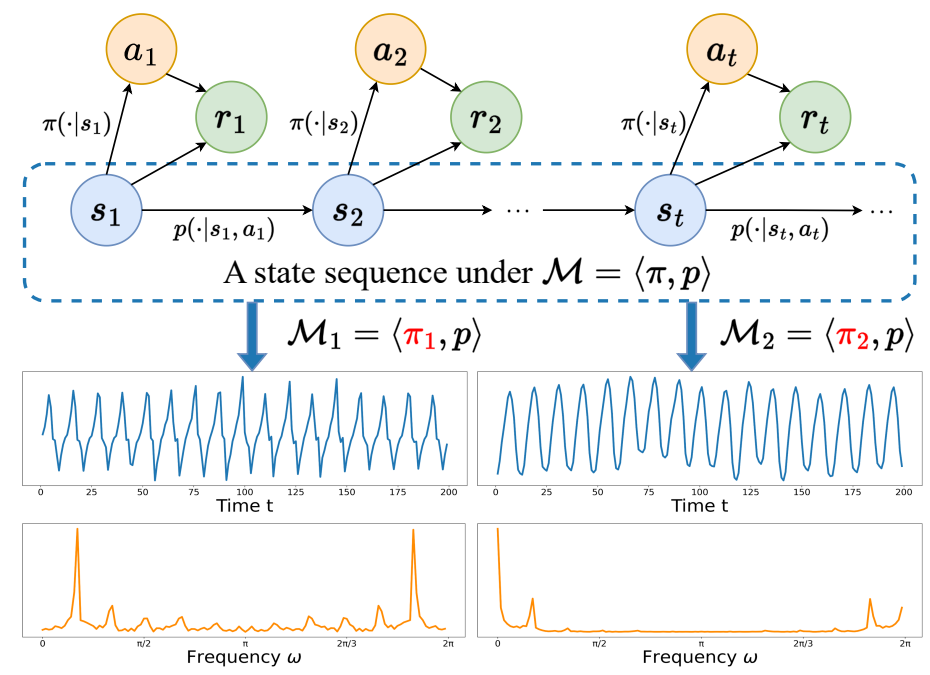

Figure 1: **State Sequence Generation Process**. The top row displays a sample of state sequences generated from MDP $\mathcal{M}$. The bottom two rows visualize two state sequences in the time and frequency domains respectively. Each column corresponds to a state sequence generated from a different policy.

Despite these benefits of state sequences, it is challenging to extract features from these sequential data through long-term prediction tasks. A substantial obstacle is the difficulty of learning accurate long-term prediction models for feature extraction. Previous methods propose making multi-step predictions using a one-step dynamic model by repeatedly feeding the prediction back into the learned model [11–13]. However, these approaches require a high degree of accuracy in the one-step model to avoid accumulating errors in multi-step predictions [13]. Another obstacle is the storage of multi-step prediction targets. For instance, some method learns a specific dynamics model to directly predict multi-step future states [14], which requires significant additional memory to store multi-step future states as prediction targets.

To tackle these problems, we propose utilizing the structural information inherent in the sequential state signals to extract useful features, thus circumventing the difficulty of learning an accurate prediction model. In Section 4, we theoretically demonstrate two types of sequential dependency structures present in state sequences. The first structure involves the dependency between reward sequences and state sequences, where the state sequences implicitly reflect the performance of the current policy and exhibit significant differences under good and bad policies. The second structure pertains to the temporal dependencies among the state signals, namely the regularity patterns exhibited by the state sequences. By exploiting the structural information, representations can focus on the underlying critical features of long-term signals, thereby reducing the need for high prediction accuracy and improving training stability [13].

Building upon our theoretical analyses, we propose **S**tate Sequences **P**rediction via **F**ourier Transform (SPF), a novel method that exploits the frequency domain of state sequences to efficiently extract the underlying structural information of long-term signals. Utilizing the frequency domain offers several advantages. Firstly, it is widely accepted that the frequency domain shows the regularity properties of the time-series data [15–17]. Secondly, we demonstrate in Section 5.1 that the Fourier transform of state sequences retains the ability to indicate policy performance under certain assumptions. Moreover, Figure 1 provides an intuitive understanding that the frequency domain enables more effective discrimination of two similar temporal signals that are difficult to differentiate in the time domain, thereby improving the efficiency of policy performance distinction and policy learning.

Specifically, our method performs an auxiliary self-supervision task that predicts the Fourier transform (FT) of infinite-step state sequences to improve the efficiency of representation learning. To facilitate the practical implementation of our method, we reformulate the Fourier transform of state sequences as a recursive form, allowing the auxiliary loss to take the form of a TD error [18], which depends only on the single-step future state. Therefore, SPF is simple to implement and eliminates the requirements for storing infinite-step state sequences when computing the labels of FT. Experiments demonstrate that our method outperforms several state-of-the-art algorithms in terms of both sample efficiency and performance on six MuJoCo tasks. Additionally, we visualize the fine distinctions between the multi-step future states recovered from our predicted FT and the true states, which indicates that our representation effectively captures the inherent structures of future state sequences.

## 2 Related Work

**Learning Predictive Representations in RL.** Many existing methods leverage auxiliary tasks that predict single-step future reward or state signals to improve the efficiency of representation learning [2, 5, 19]. However, multi-step future signals inherently contain more valuable features for long-term decision-making than single-step signals. Recent work has demonstrated the effectiveness of using future reward sequences as supervisory signals to improve the generalization performance of visual RL algorithms [9, 10]. Several studies propose making multi-step predictions of state sequences using a one-step dynamic model by repeatedly feeding the prediction back into the learned model, which is applicable to both model-free [11] and model-based [12] RL. However, these approaches require a high degree of accuracy in the one-step model to prevent accumulating errors [13]. To tackle this problem, the existing method learned a prediction model to directly predict multi-step future states [14], which results in significant additional storage for the prediction labels. In our work, we propose to predict the FT of state sequences, which reduces the demand for high prediction accuracy and eliminates the need to store multi-step future states as prediction targets.

**Incorporating the Fourier Features.** There existing many traditional RL methods to express the Fourier features. Early works have investigated representations based on a fixed basis such as Fourier basis to decompose the function into a sum of simpler periodic functions [20, 18]. Another research explored enriching the representational capacity using random Fourier features of the observations. Moreover, in the field of self-supervised pre-training, neuro2vec [16] conducts representation learning by predicting the Fourier transform of the masked part of the input signal, which is similar to our work but needs to store the entire signal as the label.

## 3 Preliminaries

### 3.1 MDP Notation

In RL tasks, the interaction between the environment and the agent is modeled as a Markov decision process (MDP). We consider the standard MDP framework [21], in which the environment is given by the tuple $\mathcal{M} := \langle \mathcal{S}, \mathcal{A}, R, P, \mu, \gamma \rangle$, where $\mathcal{S}$ is the set of states, $\mathcal{A}$ is the set of actions, $R : \mathcal{S} \times \mathcal{A} \times \mathcal{S} \to [-R_{\text{max}}, R_{\text{max}}]$ is the reward function, $P : \mathcal{S} \times \mathcal{A} \times \mathcal{S} \to [0, 1]$ is the transition probability function, $\mu : \mathcal{S} \to [0, 1]$ is the initial state distribution, and $\gamma \in [0, 1)$ is the discount factor. A policy $\pi$ defines a probability distribution over actions conditioned on the state, i.e. $\pi(a|s)$. The environment starts at an initial state $s_0 \sim \mu$. At time $t \geq 0$, the agent follows a policy $\pi$ and selects an action $a_t \sim \pi(\cdot|s_t)$. The environment then stochastically transitions to a state $s_{t+1} \sim P(\cdot|s_t, a_t)$ and produces a reward $r_t = R(s_t, a_t, s_{t+1})$. The goal of RL is to select an optimal policy $\pi^*$ that maximizes the cumulative sum of future rewards. Following previous work [22, 23], we define the *performance* of a policy $\pi$ as its expected sum of future discounted rewards:

$$J(\pi, \mathcal{M}) := E_{\tau \sim (\pi, \mathcal{M})} \left[ \sum_{t=0}^{\infty} \gamma^t R(s_t, a_t, s_{t+1}) \right], \tag{1}$$

where $\tau := (s_0, a_0, s_1, a_1, \cdots)$ denotes a trajectory generated from the interaction process and $\tau \sim (\pi, \mathcal{M})$ indicates that the distribution of $\tau$ depends on $\pi$ and the environment model $\mathcal{M}$. For simplicity, we write $J(\pi)$ and $\tau \sim \pi$ as shorthand since our environment is stationary. We also interest about the discounted future state distribution $d^\pi$, which is defined by $d^\pi(s) = (1-\gamma) \sum_{t=0}^{\infty} \gamma^t P(s_t = s|\pi, \mathcal{M})$. It allows us to express the expected discounted total reward compactly as

$$J(\pi) = \frac{1}{1-\gamma} E_{\substack{s \sim d^\pi \\ a \sim \pi \\ s' \sim P}} \left[ R(s, a, s') \right]. \tag{2}$$

The proof of (2) can be found in [24] or Section A.1 in the appendix.

Given that we aim to train an encoder that effectively captures the useful aspects of the environment, we also consider a latent MDP $\overline{\mathcal{M}} = \langle \overline{\mathcal{S}}, \mathcal{A}, \overline{R}, \overline{P}, \overline{\mu}, \gamma, \rangle$, where $\overline{\mathcal{S}} \subset \mathbb{R}^D$ for finite $D$ and the action space $\mathcal{A}$ is shared by $\mathcal{M}$ and $\overline{\mathcal{M}}$. We aim to learn an embedding function $\phi : \mathcal{S} \to \overline{\mathcal{S}}$, which connects the state spaces of these two MDPs. We similarly denote $\overline{\pi}^*$ as the optimal policy in $\overline{\mathcal{M}}$. For ease of notation, we use $\overline{\pi}(\cdot|s) := \overline{\pi}(\cdot|\phi(s))$ to represent first using $\phi$ to map $s \in \mathcal{S}$ to the latent state space $\overline{\mathcal{S}}$ and subsequently using $\overline{\pi}$ to generate the probability distribution over actions.

### 3.2 Discrete-Time Fourier Transform

The discrete-time Fourier transform (DTFT) is a powerful way to decompose a time-domain signal into different frequency components. It converts a real or complex sequence $\{x_n\}_{n=-\infty}^{+\infty}$ into a complex-valued function $F(\omega) = \sum_{n=-\infty}^{\infty} x_n e^{-j\omega n}$, where $\omega$ is a frequency variable. Due to the discrete-time nature of the original signal, the DTFT is $2\pi$-periodic with respect to its frequency variable, i.e., $F(\omega + 2\pi) = F(\omega)$. Therefore, all of our interest lies in the range $\omega \in [0, 2\pi]$ that contains all the necessary information of the infinite-horizon time series.

It is important to note that the signals considered in this paper, namely state sequences, are real-valued, which ensures the conjugate symmetry property of DTFT, i.e., $F(2\pi - \omega) = F^*(\omega)$. Therefore, in practice, it suffices to predict the DTFT only on the range of $[0, \pi]$, which can reduce the number of parameters and further save storage space.

## 4 Structural Information in State Sequences

In this section, we theoretically demonstrate the existence of the structural information inherently in the state sequences. We argue that there are two types of sequential dependency structures present in state sequences, which are useful for indicating policy performance and capturing regularity features of the states, respectively.

### 4.1 Policy Performance Distinction via State Sequences

In the RL setting, it is widely recognized that pursuing the highest reward at each time step in a greedy manner does not guarantee the maximum long-term benefit. For that reason, RL algorithms optimize the objective of cumulative reward over an episode, rather than the immediate reward, to encourage the agent to make farsighted decisions. This is why previous work has leveraged information about future reward sequences to capture long-term features for stronger representation learning [9].

Compared to the sparse reward signals, we claim that sequential state signals contain richer information. In MDP, the stochasticity of a trajectory derives from random actions selected by the agent and the environment's subsequent transitions to the next state and reward. These two sources of stochasticity are modeled as the policy $\pi(a|s)$ and the transition $p(s', r|s, a)$, respectively. Both of them are conditioned on the current state. Over long interaction periods, the dependencies of action and reward sequences on state sequences become more evident. That is, the sequence of future states largely determines the sequence of actions that the agent selects and further determines the corresponding sequence of rewards, which implies the trend and performance of the current policy, respectively. Thus, state sequences not only explicitly contain information about the environment's dynamics model, but also implicitly reveal information about policy performance.

We provide further theoretical justification for the above statement, which shows that the distribution distance between two state sequences obtained from different policies provides an upper bound on the performance difference between those policies, under certain assumptions about the reward function.

**Theorem 1.** *Suppose that the reward function $R(s, a, s') = R(s)$ is related to the state $s$, then the performance difference between two arbitrary policies $\pi_1$ and $\pi_2$ is bounded by the L1 norm of the difference between their state sequence distributions:*

$$|J(\pi_1) - J(\pi_2)| \leq \frac{R_{max}}{1-\gamma} \cdot \left\| P(s_0, s_1, s_2, \ldots | \pi_1, \mathcal{M}) - P(s_0, s_1, s_2, \ldots | \pi_2, \mathcal{M}) \right\|_1 , \quad (3)$$

*where $P(s_0, s_1, s_2, \ldots | \pi, \mathcal{M})$ means the joint distribution of the infinite-horizon state sequence $\mathbf{S} = \{\mathbf{s_0}, \mathbf{s_1}, \mathbf{s_2}, \ldots\}$ conditioned on the policy $\pi$ and the environment model $\mathcal{M}$.*

The proof of this theorem is provided in Appendix A.1. The theorem demonstrates that the greater the difference in policy performance, the greater the difference in their corresponding state sequence distributions. When we adjust the ratio $\frac{R_{max}}{1-\gamma}$ to take a relatively small value by scaling the reward, the theorem indicates that good and bad policies generate significantly different state sequence distributions. Furthermore, it confirms that learning via state sequences can significantly influence the search for policies with good performance.

### 4.2 Asymptotic Periodicity of States in MDP

Many tasks in real scenarios exhibit periodic behavior as the underlying dynamics of the environment are inherently periodic, such as industrial robots, car driving in specific scenarios, and area sweeping tasks. Take the assembly robot as an example, the robot is trained to assemble parts together to create a final product. When the robot reaches a stable policy, it executes a periodic sequence of movements that allow it to efficiently assemble the parts together. In the case of MuJoCo tasks, the agent also exhibits periodic locomotion when reaching a stable policy. We provide a corresponding video in the supplementary material to show the periodic locomotion of several MuJoCo tasks.

Inspired by these cases, we provide some theoretical analysis to demonstrate that, under some assumptions about the transition probability matrices, the state sequences in finite state space may exhibit asymptotically periodic behaviors when the agent reaches a stable policy.

**Theorem 2.** *Suppose that the state space $\mathcal{S}$ is finite with a transition probability matrix $P \in \mathbb{R}^{|\mathcal{S}|\times|\mathcal{S}|}$ and $\mathcal{S}$ has $\alpha$ recurrent classes. Let $R_1, R_2, \dots, R_\alpha$ be the probability submatrices corresponding to the recurrent classes and let $d_1, d_2, \dots, d_\alpha$ be the number of the eigenvalues of modulus 1 that the submatrices $R_1, R_2, \dots, R_\alpha$ has. Then for any initial distribution $\mu_0$, $P^n\mu_0$ is asymptotically periodic with period $d = \operatorname{lcm}(d_1, d_2, \dots, d_\alpha)$.*

The proof of the theorem is provided in Appendix A.2. The above theorem demonstrates that there exist regular and highly-structured features in the state sequences, which can be used to learn an expressive representation. Note that in an infinite state space, if the Markov chain contains a recurrent class, then after a sufficient number of steps, the state will inevitably enter one of the recurrent classes. In this scenario, the asymptotic periodicity of the state sequences can be also analyzed using the aforementioned theorem. Furthermore, even if the state sequences do not exhibit a strictly periodic pattern, regularities still exist within the sequential data that can be extracted as representations to facilitate policy learning.

## 5 Method

In the previous section, we demonstrate that state sequences contain rich structural information which implicitly indicates the policy performance and regular behavior of states. However, such information is not explicitly shown in state sequences in the time domain. In this section, we describe how to effectively leverage the inherent structural information in time-series data.

### 5.1 Learning via Frequency Domain of State Sequences

In this part, we will discuss the advantages of leveraging the frequency pattern of state sequences for capturing the inherent structural information above explicitly and efficiently.

Based on the following theorem, we find that the FT of the state sequences preserves the property in the time domain that the distribution difference between state sequences controls the performance difference between the corresponding two policies, but is subject to some stronger assumptions.

**Theorem 3.** *Suppose that $\mathcal{S} \subset \mathbb{R}^D$ the reward function $R(s, a, s') = R(s)$ is an nth-degree polynomial function with respect to $s \in \mathcal{S}$, then for any two policies $\pi_1$ and $\pi_2$, their performance difference can be bounded as follows:*

$$|J(\pi_1) - J(\pi_2)| \leq \frac{\sqrt{D}}{1-\gamma} \cdot \sum_{k=1}^{n} \frac{\left\| R^{(k)}(0) \right\|_D}{k!} \cdot \max_{1\leq i\leq D} \sup_{\omega_i\in[0,2\pi]} \left| F_{\pi_1}^{(k)}(\omega_i) - F_{\pi_2}^{(k)}(\omega_i) \right|, \tag{4}$$

*where $F_\pi^{(k)}(\omega)$ denotes the DTFT of the time series $\mathbf{S}^{(k)} = \{\mathbf{s_0}^k, \mathbf{s_1}^k, \mathbf{s_2}^k, \dots\}$ for any integer $k \in [1, n]$ and $\mathbf{S}^{(k)}$ means the kth power of the state sequence produced by the policy $\pi$. The dimensionality of $\omega$ is the same as $s$.*

We provide the proof in Appendix A.1. Similar to the analysis of Theorem 1, the above theorem shows that state sequences in the frequency domain can indicate the policy performance and can be leveraged to enhance the search for optimal policies. Furthermore, the Fourier transform can decompose the state sequence signal into multiple physically meaningful components. This operator enables the analysis of time-domain signals in a higher dimensional space, making it easier to

distinguish between two segments of signals that appear similar in the time domain. In addition, periodic signals have distinctive characteristics in their Fourier transforms due to their discrete spectra. We provide a visualization of the DTFT of the state sequences in Appendix E, which reveals that the DTFT of the periodic state sequence is approximately discrete. This observation suggests that the periodic information of the signal can be explicitly extracted in the frequency domain, particularly for the periodic cases provided by Theorem 2. For non-periodic sequences, some regularity information can still be obtained by the frequency range that the signals carry.

In addition to those advantages, the operation of the Fourier transforms also yields a concise auxiliary objective similar to the TD-error loss [18], which we will discuss in detail in the following section.

### 5.2 Learning Objective of SPF

In this part, we propose our method, **S**tate Sequences **P**rediction via **F**ourier Transform (SPF), and describe how to utilize the frequency pattern of state sequences to learn an expressive representation. Specifically, our method performs an auxiliary self-supervision task by predicting the discrete-time Fourier transform (DTFT) of infinite-step state sequences to capture the structural information in the state sequences for representation learning, hence improving upon the sample efficiency of learning.

Now we model the auxiliary self-supervision task. Given the current observation $s_t$ and the current action $a_t$, we define the expectation of future state sequence $\widetilde{s}_t$ over infinite horizon as

$$[\widetilde{s}_t]_n = [\widetilde{s}(s_t, a_t)]_n = \begin{cases} \gamma^n E_{\pi,p}\left[s_{t+n+1} \middle| s_t, a_t\right] & n = 0, 1, 2, \ldots \\ 0 & n = -1, -2, -3 \ldots \end{cases} . \tag{5}$$

Then the discrete-time Fourier transform of $\widetilde{s}_t$ is $\mathcal{F}\widetilde{s}_t(\omega) = \sum_{n=0}^{+\infty} \left[\widetilde{s}_t\right]_n e^{-j\omega n}$, where $\omega$ represents the frequency variable. The discount factor $\gamma$ in (5) is used to ensure the convergence of the Fourier transform and also serves as the contraction factor in the following Theorem 4. Since the state sequences are discrete-time signals, the corresponding DTFT is $2\pi$-periodic with respect to $\omega$. Based on this property, a common practice for operational feasibility is to compute a discrete approximation of the DTFT over one period, by sampling the DTFT at discrete points over $[0, 2\pi]$. In practice, we take $L$ equally-spaced samples of the DTFT. Then the prediction target is a matrix with size $L * D$, where $D$ is the dimension of the state space. We can derive that the DTFT functions at successive time steps are related to each other in a recursive form:

$$F_{\pi,p}(s_t, a_t) = \widetilde{\boldsymbol{S}}_t + \Gamma\, E_{\pi,p}\left[F(s_{t+1}, a_{t+1})\right]. \tag{6}$$

The detailed derivation and the specific form of $\widetilde{\boldsymbol{S}}_t$ and $\Gamma$ is provided in Appendix A.3.

Based on the recursive formula (6), we can obtain the prediction loss by computing the difference between the estimated Fourier value $F_{\pi,p}(s_t, a_t)$ and the better estimate $\widetilde{\boldsymbol{S}}_t + \Gamma\, E_{\pi,p}\left[F(s_{t+1}, a_{t+1})\right]$, just like the TD error. Similar to the TD-learning of value functions, the recursive relationship can be reformulated as contraction mapping $\mathcal{T}$, as shown in the following theorem (see proof in Appendix A.3). Due to the properties of contraction mappings, we can iteratively apply the operator $\mathcal{T}$ to compute the target DTFT function of long-term state sequences until convergence in tabular settings. When calculating the prediction loss, we only need to utilize the current state $s_t$, the current action $a_t$, and the next state $s_{t+1}$. Therefore, one notable advantage of SPF is that there is no need to store multi-step future states as labels for predicting future state sequences.

**Theorem 4.** *Let $\mathcal{F}$ denote the set of all functions $F : \mathcal{S} \times \mathcal{A} \to \mathbb{C}^{L*D}$ and define the norm on $\mathcal{F}$ as*

$$\|F\|_{\mathcal{F}} := \sup_{\substack{s\in\mathcal{S} \\ a\in\mathcal{A}}} \max_{0\le k<L} \left\| \left[F(s,a)\right]_k \right\|_D ,$$

*where $\left[F(s,a)\right]_k$ represents the kth row vector of $F(s,a)$. We show that the mapping $\mathcal{T} : \mathcal{F} \to \mathcal{F}$ defined as*

$$\mathcal{T}F(s_t, a_t) = \widetilde{\boldsymbol{S}}_t + \Gamma\, E_{\pi,p}\left[F(s_{t+1}, a_{t+1})\right] \tag{7}$$

*is a contraction mapping, where $\widetilde{\boldsymbol{S}}_t$ and $\Gamma$ are defined in Appendix A.3.*

As the Fourier transform of the real state signals has the property of conjugate symmetry, we only need to predict the DTFT on a half-period interval $[0, \pi]$. Therefore, we reduce the row size of the

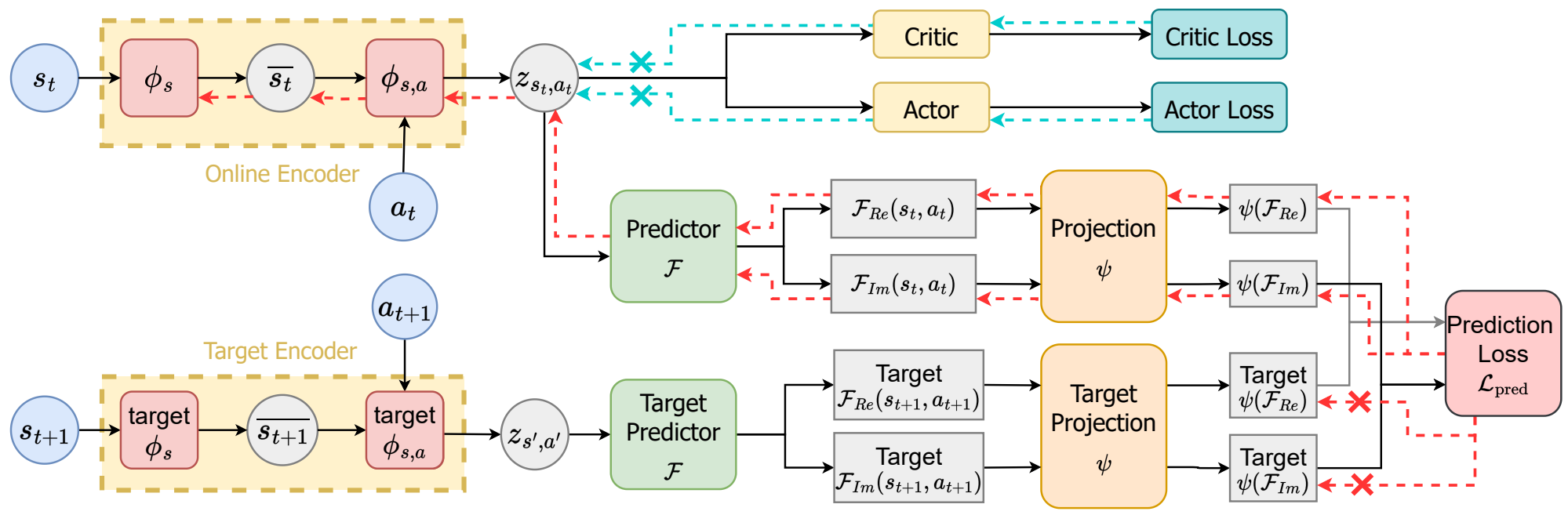

Figure 2: **The network architecture of SPF**. The online encoder $\phi$ outputs the representations used in the RL task and the predictor $\mathcal{F}$ predicts complex-valued Fourier transform of the state sequences starting from the state-action pair $(s_t, a_t)$. During training, $(s_t, a_t, s_{t+1})$ are previously experienced states and actions sampled from a replay buffer. The dashed line show how gradients flow back to model weights. We prevent the gradient of RL losses from updating the online encoder and prevent the gradient of prediction loss from updating the target encoder.

prediction target to half for reducing redundant information and saving storage space. In practice, we train a parameterized prediction model $\mathcal{F}$ to predict the DTFT of state sequences. Note that the value of the prediction target is on the complex plane, so the prediction network employs two separate output modules $\mathcal{F}_{\text{Re}}$ and $\mathcal{F}_{\text{Im}}$ as real and imaginary parts respectively. Then we define the auxiliary prediction loss function as:

$$L_{\text{pred}}(\phi, \mathcal{F}) = d\left(\widetilde{\boldsymbol{S}}_t + \left[\Gamma_{\text{Re}}\mathcal{F}_{\text{Re}}(\overline{s_{t+1}}, \pi(\overline{s_{t+1}})) - \Gamma_{\text{Im}}\mathcal{F}_{\text{Im}}(\overline{s_{t+1}}, \pi(\overline{s_{t+1}}))\right],\ \mathcal{F}_{\text{Re}}(\overline{s_t}, a_t)\right) + d\Big(\left[\Gamma_{\text{Im}}\mathcal{F}_{\text{Re}}(\overline{s_{t+1}}, \pi(\overline{s_{t+1}})) + \Gamma_{\text{Re}}\mathcal{F}_{\text{Im}}(\overline{s_{t+1}}, \pi(\overline{s_{t+1}}))\right],\ \mathcal{F}_{\text{Im}}(\overline{s_t}, a_t)\Big),$$

where $\overline{s_t} = \phi(s_t)$ means the representation of the state, $\Gamma_{\text{Re}}$ and $\Gamma_{\text{Im}}$ denote the real and imaginary parts of the constant $\Gamma$, and $d$ denotes an arbitrary similarity measure. We choose $d$ as cosine similarity in practice. The algorithm pseudo-code is shown in Appendix B.

### 5.3 Network Architecture of SPF

Here we provide more details about the network architecture of our method. In addition to the encoder and the predictor, we use projection heads $\psi$ [25] that project the predicted values onto a low-dimensional space to prevent overfitting when computing the prediction loss directly from the high-dimensional predicted values. In practice, we use a loss function called *freqloss*, which preserves the low and high-frequency components of the predicted DTFT without the dimensionality reduction process (See Appendix C for more details). Furthermore, when computing the target predicted value, we follow prior work [26, 11] to use the target encoder, predictor, and projection for more stable performance. We periodically overwrite the target network parameters with an exponential moving average of the online network parameters.

In the training process, we train the encoder $\phi$, the predictor $\mathcal{F}$, and the projection $\psi$ to minimize the auxiliary prediction loss $\mathcal{L}_{\text{pred}}(\phi, \mathcal{F}, \psi)$, and alternately update the actor-critic models of RL tasks using the trained encoder $\phi$. We illustrate the overall architecture of SPF and the gradient flows during training in Figure 2.

## 6 Experiments

We quantitatively evaluate our method on a standard continuous control benchmark—the set of MuJoCo [27] environments implemented in OpenAI Gym.

### 6.1 Comparative Evaluation on MuJoCo

To evaluate the effect of learned representations, we measure the performance of two traditional RL methods SAC [28] and PPO [29] with raw states, OFENet representations [19], and SPF represen-

Table 1: **Mean and standard error results on six MuJoCo tasks at 500K step and 1M step**. All means and standard errors are calculated over 10 seeds. The highest mean scores are marked in blue. SPF outperforms other SOTA methods in 5 out of 6 settings with an average 19.5% boost.

| | Environment | **SAC-SPF** | SAC-OFE | SAC-raw | **PPO-SPF** | PPO-OFE | PPO-raw |
|---|---|---|---|---|---|---|---|
| 500K step | HalfCheetah | $\mathbf{12739 \pm 139}$ $(+7\%)$ | $11932 \pm 75$ | $9004 \pm 150$ | $\mathbf{2448 \pm 357}$ $(+1\%)$ | $2419 \pm 296$ | $1475 \pm 238$ |
| | Hopper | $\mathbf{3443 \pm 24}$ $(+15\%)$ | $2983 \pm 123$ | $2343 \pm 183$ | $\mathbf{2923 \pm 98}$ $(+36\%)$ | $1529 \pm 157$ | $2156 \pm 245$ |
| | Walker2d | $\mathbf{4868 \pm 53}$ $(+29\%)$ | $3762 \pm 162$ | $2023 \pm 269$ | $\mathbf{1203 \pm 211}$ $(+45\%)$ | $339 \pm 32$ | $827 \pm 156$ |
| | Ant | $\mathbf{6531 \pm 175}$ $(+17\%)$ | $5587 \pm 253$ | $2691 \pm 76$ | $867 \pm 34$ | $696 \pm 47$ | $\mathbf{929 \pm 15}$ |
| | Swimmer | $45 \pm 0$ | $45 \pm 0$ | $42 \pm 0$ | $\mathbf{90 \pm 11}$ $(+15\%)$ | $78 \pm 12$ | $66 \pm 6$ |
| | Humanoid | $\mathbf{4613 \pm 118}$ $(+13\%)$ | $4071 \pm 172$ | $2953 \pm 136$ | $395 \pm 10$ | $\mathbf{405 \pm 18}$ | $383 \pm 9$ |
| 1M step | HalfCheetah | $\mathbf{15822 \pm 109}$ $(+10\%)$ | $14425 \pm 112$ | $10745 \pm 159$ | $\mathbf{3154 \pm 1358}$ $(+3\%)$ | $3066 \pm 326$ | $2259 \pm 344$ |
| | Hopper | $\mathbf{3517 \pm 20}$ $(+10\%)$ | $3197 \pm 147$ | $3056 \pm 91$ | $\mathbf{3152 \pm 137}$ $(+16\%)$ | $2370 \pm 258$ | $2721 \pm 222$ |
| | Walker2d | $\mathbf{5158 \pm 69}$ $(+7\%)$ | $4833 \pm 59$ | $3367 \pm 110$ | $2229 \pm 299$ | $1080 \pm 236$ | $\mathbf{2302 \pm 158}$ |
| | Ant | $\mathbf{7241 \pm 85}$ $(+7\%)$ | $6738 \pm 150$ | $4220 \pm 164$ | $\mathbf{1053 \pm 79}$ $(+15\%)$ | $913 \pm 18$ | $867 \pm 17$ |
| | Swimmer | $45 \pm 0$ | $45 \pm 0$ | $43 \pm 0$ | $\mathbf{99 \pm 13}$ $(+19\%)$ | $73 \pm 10$ | $83 \pm 9$ |
| | Humanoid | $5633 \pm 112$ | $\mathbf{6241 \pm 98}$ | $4618 \pm 118$ | $441 \pm 17$ | $\mathbf{448 \pm 17}$ | $439 \pm 18$ |

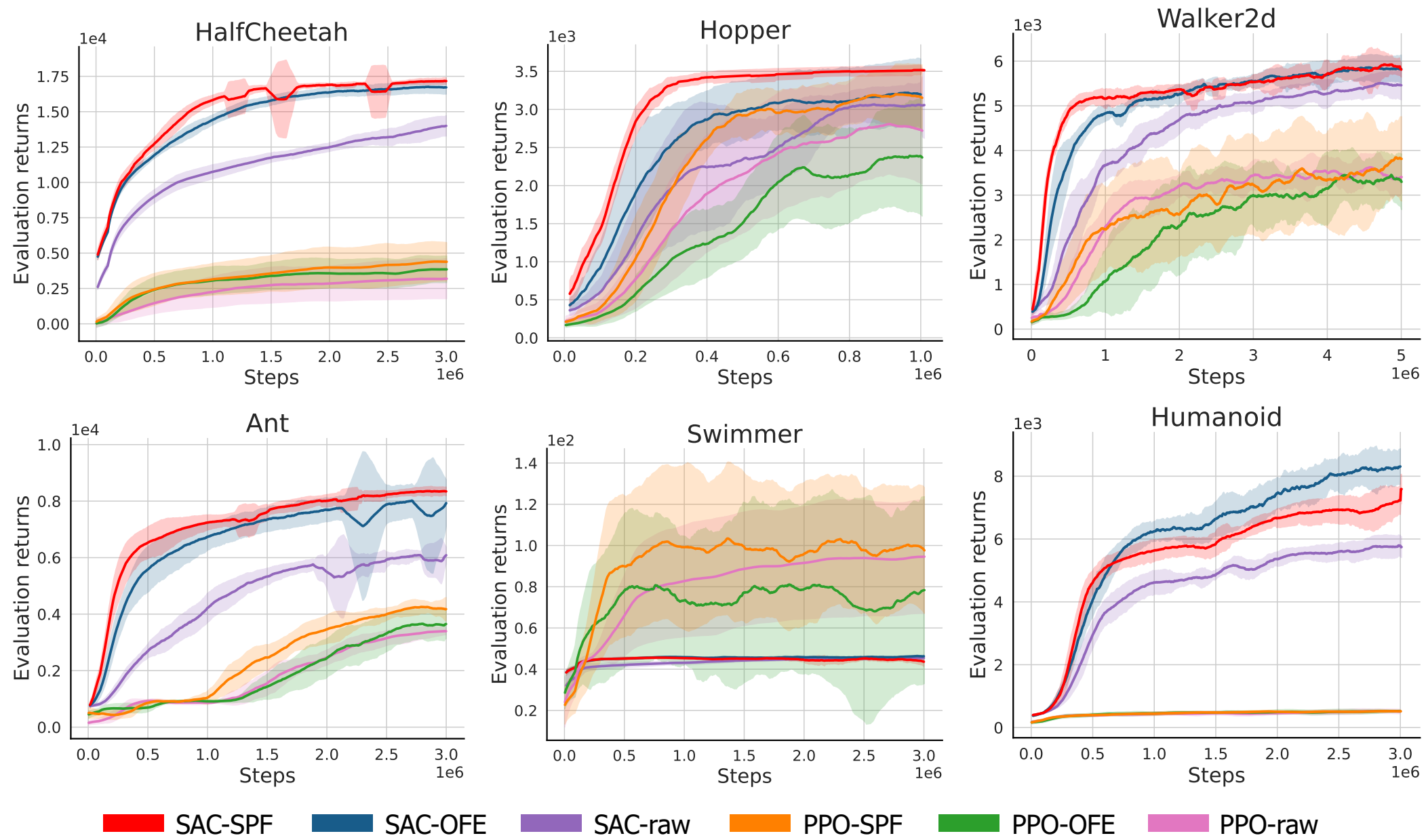

Figure 3: **Results on six MuJoCo tasks**. The solid curves denote the means and the shaded regions denote the minimum and maximum returns over 10 seeds. Each checkpoint is evaluated by 10 episodes in evaluated environments. Curves are smoothed for visual clarity.

tations. We train our representations via the Fourier transform of infinite-horizon state sequences, which means the representations learn to capture the information of infinite-step future states. Hence, we compare our representation with OFENet, which uses the auxiliary task of predicting single-step future states for representation learning. For a fair comparison, we use an encoder with the same network structure as that of OFENet. See Appendix C and D for more details about network architectures and hyperparameters setting.

As described above, our comparable methods include: 1) **SAC-SPF**: SAC with SPF representations; 2) **SAC-OFE**: SAC with OFENet representations; 3) **SAC-raw**: SAC with raw states; 4) **PPO-SPF**: PPO with SPF representations; 5) **PPO-OFE**: PPO with OFENet representations; 6) **PPO-raw**: PPO with raw states. Figure 3 shows the learning curves of the above methods. As our results suggest, SPF shows superior performance compared to the original algorithms, SAC-raw and PPO-raw, across all six MuJoCo tasks and also outperforms OFENet in terms of both sample efficiency and asymptotic performance on five out of six MuJoCo tasks. According to the results in Table 1, SPF achieves an average gain of +19.5% over other methods at 500K step and 1M step. These results indicate that our approach learns more quickly than other methods when the number of interactions is limited. One possible explanation for the comparatively weaker performance of our method on the Humanoid-v2

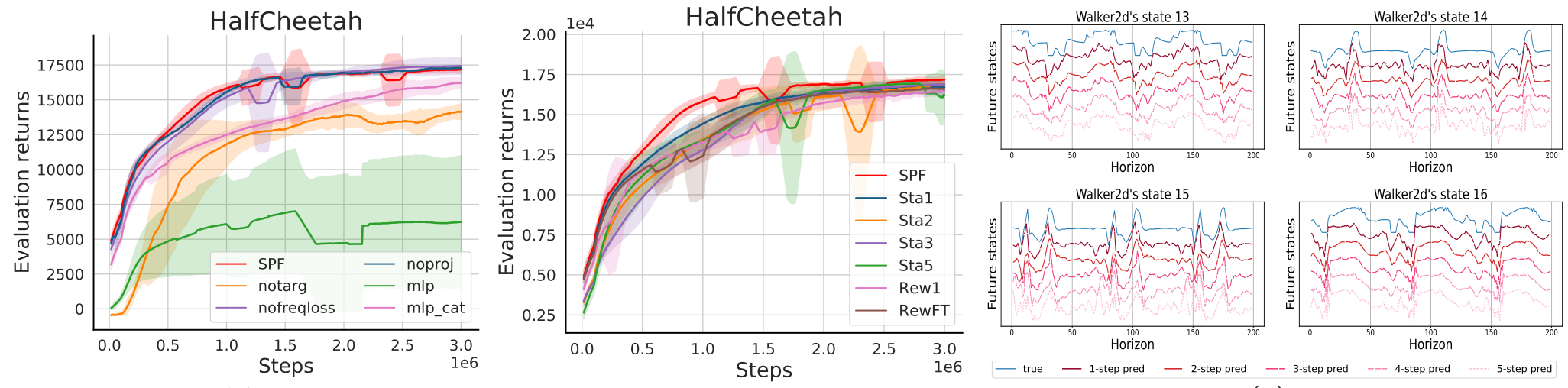

(a) (b) (c)

Figure 4: **Results of additional trials**. **(a)** Ablation study: training curves of five variant methods of SPF on SAC. Each method is evaluated over 5 seeds on HalfCheetah-v2. **(b)** Comparison of prediction targets: training curves of SAC with different auxiliary prediction tasks. Each method is evaluated over 5 seeds on HalfCheetah-v2. **(c)** Visualization of recovered states from the predicted DTFT. The blue line represents the true state sequence, while the red line represents the recovered state sequence. The lighter red line corresponds to predictions made by historical states from a more distant time step. The four subfigures represent four dimensions of the state space on Walker2d.

task is that its state contains the external forces to bodies, which show limited regularities due to their discontinuous and sparse.

### 6.2 Ablation Study

In this part, we will verify that just predicting the FT of state sequences may fall short of the expected performance and that using SPF is necessary to get better performance. To this end, we conducted an ablation study to identify the specific components that contribute to the performance improvements achieved by SPF. Figure 4(a) shows the ablation study over SAC with HalfCheetah-v2 environment.

*notarg* removes all target networks of the encoder, predictor, and projection layer from SPF. Based on the results, the variant of SPF exhibits significantly reduced performance when target estimations in the auxiliary loss are generated by the online encoder without a stopgradient. Therefore, using a separate target encoder is vital, which can significantly improve the stability and convergence properties of our algorithm.

*nofreqloss* computes the cosine similarity of the projection layer's outputs directly without any special treatment to the low-frequency and high-frequency components of our predicted DTFT. The reduced convergence rate of *nofreqloss* suggests that preserving the complete information of low and high-frequency components can encourage the representations to capture more structural information in the frequency domain.

*noproj* removes the projection layer from SPF and computes the cosine similarity of the predicted values as the objective. The performance did not significantly deteriorate after removing the projection layer, which indicates that the prediction accuracy of state sequences in the frequency domain may not have a strong impact on the quality of representation learning. Therefore, it can be inferred that SPF places a greater emphasis on capturing the underlying structural information of the state sequences, and is capable of reconstructing the state sequences with a low risk of overfitting.

*mlp* changes the lay block of the encoder from MLP-DenseNet to MLP. The much lower scores of *mlp* indicate that both the raw state and the output of hidden layers contain important information that contributes to the quality of the learned representations. This result underscores the importance of leveraging sufficient information for representation learning.

*mlp-cat* uses a modified block of MLP as the layer of the encoder, which concat the output of MLP with the raw state. The performance of *mlp-cat* does increase compared to *mlp*, but is still not as good as SPF in terms of both sample efficiency and performance.

### 6.3 Comparison of Different Prediction Targets

This section aims to test the effect of our prediction target—infinite-step state sequences in the frequency domain—on the efficiency of representation learning. We test five types of prediction

targets: 1) **Sta1**: single-step future state; 2) **StaN**: N-step state sequences, where we choose $N = 2, 3, 5$; 3) **SPF**: infinite-step state sequences in frequency domain; 4) **Rew1**: single-step future reward; 5) **RewFT**: infinite-step reward sequences in frequency domain.

As shown in Figure 4(b), SPF outperforms all other competitors in terms of sample efficiency, which indicates that infinite-step state sequences in the frequency domain contain more underlying valuable information that can facilitate efficient representation learning. Since Sta1 and SPF outperform Rew1 and RewFT respectively, it can be referred that learning via states is more effective for representation learning than learning via rewards. Notably, the lower performance of StaN compared to Sta1 could be attributed to the model's tendency to prioritize prediction accuracy over capturing the underlying structured information in the sequential data, which may impede its overall learning efficiency.

### 6.4 Visualization of Recovered State Sequences

This section aims to demonstrate that the representations learned by SPF effectively capture the structural information contained in infinite-step state sequences. To this end, we compare the true state sequences with the states recovered from the predicted DTFT via the inverse DTFT (See Appendix E for more implementation details). Figure 4(c) shows that the learned representations can recover the true state sequences even using the historical states that are far from the current time step. In Appendix E, we also provide a visualization of the predicted DTFT, which is less accurate than the results in Figure 4(c). Those results highlight the ability of SPF to effectively extract the underlying structural information in infinite-step state sequences without relying on high prediction accuracy.

We further provide a comparison table that measures the distance between the real DTFT and the predicted DTFT using cosine similarity. The results provided in Appendix F indicate that the prediction module $\mathcal{F}$ exhibits moderate predictive accuracy in approximating the real Fourier transform, with an average cosine similarity value of $-0.6$.

## 7 Conclusion

In this paper, we theoretically analyzed the existence of structural information in state sequences, which is closely related to policy performance and signal regularity, and then introduced State Sequences Prediction via Fourier Transform (SPF), a representation learning method that predicts the FT of state sequences to extract the underlying structural information in state sequences for learning expressive representations efficiently. SPF outperforms several state-of-the-art algorithms in terms of both sample efficiency and performance. Our additional experiments and visualization show that SPF encourages representations to place a greater emphasis on capturing the underlying pattern of time-series data, rather than pursuing high accuracy of prediction tasks.

**Limitations** One of the main limitations of our paper is that we have only evaluated our method on tasks where the state sequences exhibit strong asymptotic periodicity. Considering that the Fourier transform converts non-periodic signals into a continuous frequency domain, it is more difficult to extract meaningful frequency features of non-periodic signals compared to periodic signals that have a discrete frequency domain. Moreover, the frequency features extracted by our approach inherently depend on the policy and task, which limits their reusability across multiple tasks. Further research is needed to analyze the applicability of our approach to non-periodic tasks and the potential for generalization across multiple tasks.

## Acknowledgments

The authors would like to thank all the anonymous reviewers for their insightful comments. This work was supported by National Key R&D Program of China under contract 2022ZD0119801, National Nature Science Foundations of China grants U19B2026, U19B2044, 61836011, 62021001, and 61836006.

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

# A Proof

## A.1 Proof of Performance Difference Distinction via State Sequences

Following the previous work [22], our analysis will make use of the discounted future state distribution, $d^\pi$, which is defined as

$$d^\pi(s) = (1-\gamma)\sum_{t=0}^{\infty}\gamma^t P(s_t = s|\pi, \mathcal{M})$$

It allows us to express the expected discounted total reward compactly as

$$\begin{aligned}
J(\pi) &= \sum_{t=0}^{\infty}\gamma^t E_{s_t,a_t,s_{t+1}}\left[R(s_t,a_t,s_{t+1})|\pi,\mathcal{M}\right] \\
&= \sum_{t=0}^{\infty}\gamma^t \int_{\mathcal{S}} R_\pi(s)P(s_t = s|\pi,\mathcal{M})\,\mathrm{d}s \\
&= \int_{\mathcal{S}} R_\pi(s)\sum_{t=0}^{\infty}\gamma^t P(s_t = s|\pi,\mathcal{M})\,\mathrm{d}s \qquad (1) \\
&= \frac{1}{1-\gamma}\int_{\mathcal{S}} R_\pi(s)d_\pi(s)\,\mathrm{d}s \\
&= \frac{1}{1-\gamma}E_{\substack{s\sim d^\pi \\ a\sim\pi \\ s'\sim P}}\left[R(s,a,s')\right], \qquad (2)
\end{aligned}$$

where we define $R_\pi(s) := E_{a\sim\pi,s'\sim P}[R(s,a,s')]$. It should be clear from $a \sim \pi(\cdot|s)$ and $s' \sim P(\cdot|s,a)$ that $a$ and $s'$ depend on $s$. Thus, the reward function $R_\pi$ is only related to $s$ when the policy $\pi$ is fixed.

Firstly, we prove that the distance between two state sequence distributions obtained from two distinct policies serves as an upper bound on the performance difference between those policies, provided that certain assumptions regarding the reward function hold.

**Theorem 1.** *Suppose that the reward function $R(s,a,s') = R(s)$ is related to the state $s$, then the performance difference between two arbitrary policies $\pi_1$ and $\pi_2$ is bounded by the L1 norm of the difference between their state sequence distributions:*

$$|J(\pi_1) - J(\pi_2)| \leq \frac{R_{max}}{1-\gamma}\cdot \left\|P(s_0,s_1,s_2,\ldots|\pi_1,\mathcal{M}) - P(s_0,s_1,s_2,\ldots|\pi_2,\mathcal{M})\right\|_1, \qquad (3)$$

*where $P(s_0,s_1,s_2,\ldots|\pi,\mathcal{M})$ means the joint distribution of the infinite-horizon state sequence $\mathbf{S} = \{\mathbf{s_0},\mathbf{s_1},\mathbf{s_2},\ldots\}$ conditioned on the policy $\pi$ and the environment model $\mathcal{M}$.*

*Proof.* According to the equation (1), the difference in performance between two policies $\pi_1$, $\pi_2$ can be bounded as follows.

$$\begin{aligned}
|J(\pi_1) - J(\pi_2)| &\leq R_{\text{max}}\cdot\sum_{t=0}^{\infty}\gamma^t\int_{\mathcal{S}}\left|P(\mathbf{s_t} = s|\pi_1,\mathcal{M}) - P(\mathbf{s_t} = s|\pi_2,\mathcal{M})\right|\mathrm{d}s \\
&\leq R_{\text{max}}\cdot\sum_{t=0}^{T}\gamma^t\int_{\mathcal{S}}\left|\int_{\mathcal{S}^T} P(s_0,\ldots,s_{t-1},s,s_{t+1},\ldots,s_T|\pi_1,\mathcal{M})\right. \\
&\quad \left. - P(s_0,\ldots,s_{t-1},s,s_{t+1}\ldots,s_T|\pi_2,\mathcal{M})\,\mathrm{d}s_0\cdots\mathrm{d}s_{t-1}\mathrm{d}s_{t+1}\cdots\mathrm{d}s_T\right|\mathrm{d}s \\
&\quad + R_{\text{max}}\cdot 2\sum_{t=T+1}^{\infty}\gamma^t, \quad \forall T\geq 1
\end{aligned}$$

$$
\begin{aligned}
&\leq R_{\text{max}} \sum_{t=0}^{T} \gamma^t \int_{\mathcal{S}^{T+1}} \left| P(s_0, \ldots, s_T | \pi_1, \mathcal{M}) - P(s_0 \ldots, s_T | \pi_2, \mathcal{M}) \right| \mathrm{d}s_0 \cdots \mathrm{d}s_T \\
&\quad + R_{\text{max}} \cdot 2 \sum_{t=T+1}^{\infty} \gamma^t, \quad \forall T \geq 1 \\
&= \frac{R_{\text{max}}}{1-\gamma} \cdot \int_{\mathcal{S}^{T+1}} \left| P(s_0, \ldots, s_T | \pi_1, \mathcal{M}) - P(s_0 \ldots, s_T | \pi_2, \mathcal{M}) \right| \mathrm{d}s_0 \cdots \mathrm{d}s_T \\
&\quad + R_{\text{max}} \cdot 2 \sum_{t=T+1}^{\infty} \gamma^t, \quad \forall T \geq 1.
\end{aligned}
$$

Let $T \to \infty$, then we obtain the bound proposed by (3). □

We are further interested in bounding the performance difference between two policies by their state sequences in the frequency domain. Benefiting from the properties of the discrete-time Fourier transform (DTFT), we can constrain the performance difference using the Fourier transform over the interval $[0, 2\pi]$, instead of using the distribution functions of the state sequences in unbounded space.

**Theorem 2.** *Suppose that $\mathcal{S} \subset \mathbb{R}^D$ the reward function $R(s, a, s') = R(s)$ is an nth-degree polynomial function with respect to $s \in \mathcal{S}$, then for any two policies $\pi_1$ and $\pi_2$, their performance difference can be bounded as follows:*

$$
|J(\pi_1) - J(\pi_2)| \leq \frac{\sqrt{D}}{1-\gamma} \cdot \sum_{k=1}^{n} \frac{\left\| R^{(k)}(0) \right\|_D}{k!} \cdot \max_{1 \leq i \leq D} \sup_{\omega_i \in [0, 2\pi]} \left| F_{\pi_1}^{(k)}(\omega_i) - F_{\pi_2}^{(k)}(\omega_i) \right|, \tag{4}
$$

*where $F_\pi^{(k)}(\omega)$ denotes the DTFT of the time series $\mathbf{S}^{(k)} = \{\mathbf{s_0}^k, \mathbf{s_1}^k, \mathbf{s_2}^k, \ldots\}$ for any integer $k \in [1, n]$ and $\mathbf{S}^{(k)}$ means the kth power of the state sequence produced by the policy $\pi$. The dimensionality of $\omega$ is the same as $s$.*

*Proof.* For sake of simplicity, we define $p_t(s|\pi_i) = P(\mathbf{s_t} = s | \pi_i, \mathcal{M})$ for $i = 1, 2$. We denote $\varepsilon_t$ as

$$
\varepsilon_t = \int_{\mathcal{S}} R(s) \left[ p_t(s|\pi_1) - p_t(s|\pi_2) \right] \mathrm{d}s. \tag{5}
$$

Based on the Taylor series expansion, we can rewrite the reward function as $R(s) = \sum_{k=0}^{n} \frac{R^{(k)}(0)^{\mathsf{T}}}{k!} s^k$, then for any integer $k \in [1, n]$, we have

$$
\begin{aligned}
|\varepsilon_t| &\leq \sum_{k=0}^{n} \frac{\left\| R^{(k)}(0) \right\|_D}{k!} \cdot \left\| \int_{\mathcal{S}} \left[ s^k p_t(s|\pi_1) - s^k p_t(s|\pi_2) \right] \mathrm{d}s \right\|_D \\
&= \sum_{k=0}^{n} \frac{\left\| R^{(k)}(0) \right\|_D}{k!} \left\| \underset{s \sim p_t(\cdot|\pi_1)}{E} \left[ s^k \right] - \underset{s \sim p_t(\cdot|\pi_2)}{E} \left[ s^k \right] \right\|_D .
\end{aligned} \tag{6}
$$

Since the inverse DTFT of $F_\pi^{(k)}(\omega)$ is the original time series $\mathbf{S}^{(k)}$, we have

$$
\underset{s_i \sim p_t(\cdot|\pi)}{E} \left[ s_i^k \right] = \frac{1}{2\pi} \int_0^{2\pi} F_\pi^{(k)}(\omega_i) e^{j\omega_i t} \, \mathrm{d}\omega_i, \quad \forall i = 1, 2, \ldots, D. \tag{7}
$$

Then we have

$$
\begin{aligned}
\left| \underset{s_i \sim p_t(\cdot|\pi_1)}{E} \left[ s_i^k \right] - \underset{s_i \sim p_t(\cdot|\pi_2)}{E} \left[ s_i^k \right] \right| &\leq \frac{1}{2\pi} \int_0^{2\pi} \left| F_{\pi_1}^{(k)}(\omega_i) - F_{\pi_2}^{(k)}(\omega_i) \right| \cdot \left| e^{j\omega_i t} \right| \mathrm{d}\omega_i \\
&\leq \sup_{\omega_i \in [0, 2\pi]} \left| F_{\pi_1}^{(k)}(\omega_i) - F_{\pi_2}^{(k)}(\omega_i) \right| .
\end{aligned} \tag{8}
$$

Substituting (8) into (6), then we obtain

$$
|\varepsilon_t| \leq \sqrt{D} \cdot \sum_{k=1}^{n} \frac{\left\| R^{(k)}(0) \right\|_D}{k!} \cdot \max_{1 \leq i \leq D} \sup_{\omega_i \in [0, 2\pi]} \left| F_{\pi_1}^{(k)}(\omega_i) - F_{\pi_2}^{(k)}(\omega_i) \right| .
$$

For the sake of DTFT, the upper bound of $\epsilon_t$ is independent of $t$, then we could derive the performance difference bound as follows.

$$|J(\pi_1) - J(\pi_2)| \le \sum_{t=0}^{\infty} \gamma^t \cdot |\varepsilon_t|$$
$$\le \frac{1}{1-\gamma} \cdot \sqrt{D} \cdot \sum_{k=1}^{n} \frac{\left\|R^{(k)}(0)\right\|_D}{k!} \cdot \max_{1\le i\le D} \sup_{\omega_i \in [0,2\pi]} \left| F_{\pi_1}^{(k)}(\omega_i) - F_{\pi_2}^{(k)}(\omega_i) \right|,$$

and so we immediately achieve the desired bound in (4). □

### A.2 Proof of the Asymptotic Periodicity of States in MDP

This section focuses on analyzing the asymptotic behavior of the state sequences generated from an MDP. We begin by discussing the limiting process of MDP with a finite state space $\mathcal{S}$. Let $P$ be the transition probability matrix and let $\mu_i$ be the probability distribution of the states at time $t_i$. Then we have $\mu_{i+1} = P\mu_i$ for any $i \ge 0$. If the sequence $\{\mu_i\}_{i=0}^{\infty}$ splits into $d$ subsequences with $d$ cyclic limits $\{\mu_\infty^r\}_{r=0}^{d-1}$ that follow the cycle:

$$\mu_\infty^0 \to \mu_\infty^1 \to \cdots \to \mu_\infty^{d-1} \to \mu_\infty^0,$$

then we say that the states of the MDP exhibit *asymptotic periodicity*. Such cyclic asymptotic behavior implies that the limiting distribution of the states eventually repeats in a specific period after a certain number of steps.

We begin by providing some essential definitions in the field of stochastic processes [30], which will be utilized in the following proof. Let $P$ be a transition probability matrix corresponding to $n$ states ($n \ge 1$). Two states $i$ and $j$ are said to *intercommunicate* if there exist paths from $i$ to $j$ as well as from $j$ to $i$. The matrix $P$ is called *irreducible* if any two states intercommunicate. A set of states is called *irreducible* if any two states in the set intercommunicate. Moreover, a state $i$ is called *recurrent* if the probability of eventual return to $i$, having started from $i$, is $1$. If this probability is strictly less than $1$, the state $i$ is called *transient*.

Note that if the whole state space $\mathcal{S}$ is irreducible, then its transition matrix $P$ is also irreducible. The following lemma demonstrates that if the state space is irreducible, then its asymptotical periodicity is determined by the eigenvalues with modulus $1$ of its transition matrix.

**Lemma 1.** *Suppose that the state space $\mathcal{S}$ is finite with a transition probability matrix $P \in \mathbb{R}^{|\mathcal{S}|\times|\mathcal{S}|}$. If $P$ is an irreducible matrix with $d$ eigenvalues of modulus $1$, then for any initial distribution $\mu_0$, $P^n\mu_0$ is asymptotically periodic with a period of $d$ when $d > 1$ and asymptotically aperiodic when $d = 1$.*

*Proof.* According to the Perron-Frobenius theorem for irreducible non-negative matrices, all eigenvalues of $P$ of modulus $1$ are exactly the $d$ complex roots of the equation $\lambda^d - 1 = 0$. They can be formulated as $\lambda_0 = 1, \lambda_1 = \xi^1, \ldots, \lambda_{d-1} = \xi^{d-1}$, where $\xi = e^{\frac{2\pi j}{d}}$. Each of them is a simple root of the characteristic polynomial of the matrix $P$. Since $P$ is a transition probability matrix, the remaining eigenvalues $\lambda_d, \ldots, \lambda_s$ satisfy $|\lambda_r| < 1$. Therefore, the *Jordan* matrix of $P$ has the form

$$J = \begin{bmatrix} \lambda_0 & & & & & & \\ & \lambda_1 & & & & & \\ & & \ddots & & & & \\ & & & \lambda_{d-1} & & & \\ & & & & J_d & & \\ & & & & & \ddots & \\ & & & & & & J_s \end{bmatrix}, \text{where } J_k = \begin{bmatrix} \lambda_k & 1 & & & \\ & \lambda_k & 1 & & \\ & & \ddots & \ddots & \\ & & & \lambda_k & 1 \\ & & & & \lambda_k \end{bmatrix}.$$

We refer to $J_k$ as *Jordan cells*.

Let $|S| = D$, we can rewrite $P$ in its Jordan canonical form $P = XJX^{-1}$ where

$$X = [\vec{x}_0, \vec{x}_1, \ldots, \vec{x}_{D-1}].$$

Note that for $k < d$, $x_k$ is the eigenvector corresponding to $\lambda_k$. Since the column vectors of $X$ are linearly dependent, there exist $\vec{c} = [c_0, c_1, \ldots, c_{D-1}]$ not all zero, such that $\mu_0 = \sum_{k=0}^{D-1} c_k \vec{x}_k = X\vec{c}$. Thus, we have

$$P^n \mu_0 = \sum_{k=0}^{d-1} c_k \lambda_k^n \vec{x}_k + \sum_{k=d}^{D-1} c_k P^n \vec{x}_k. \tag{9}$$

For any Jordan cell $J_k$, let $\alpha_k$ be the multiplicity of $\lambda_k$, then

$$J_k^n = \begin{bmatrix} \lambda_k & 1 & & & \\ & \lambda_k & 1 & & \\ & & \ddots & \ddots & \\ & & & \lambda_k & 1 \\ & & & & \lambda_k \end{bmatrix}_{\alpha_k \times \alpha_k}^n = \begin{bmatrix} \lambda_k^n & C_n^{n-1}\lambda_k^{n-1} & \cdots & C_n^{n-\alpha_k+1}\lambda_k^{n-\alpha_k+1} \\ & \lambda_k^n & \cdots & C_n^{n-\alpha_k+2}\lambda_k^{n-\alpha_k+2} \\ & & \ddots & \vdots \\ & & & \lambda_k^n \end{bmatrix}.$$

Since $\alpha_k$ is fixed for matrix $P$, we have $\lim_{n\to\infty} J_k^n = \mathbf{0}$ for each $k = d, \ldots, D-1$. Then the limiting vector of (9), denoted by $P^\infty \mu_0$, satisfies:

$$P^\infty \mu_0 = \lim_{n\to\infty} XJ^nX^{-1}X\vec{c} = \lim_{n\to\infty} \sum_{k=0}^{d-1} c_k \lambda_k^n \vec{x}_k = \lim_{n\to\infty} \mu^{(n)},$$

where we denote $\mu^{(n)} = \sum_{k=0}^{d-1} c_k (e^{j\frac{2\pi k}{d}})^n \vec{x}_k$. Let $r = n \pmod d$, then we have

$$\mu^{(n)} = \mu^{(r)} = \sum_{k=0}^{d-1} c_k (\xi^k)^r \vec{x}_k, \quad \forall n \geq 1.$$

Therefore, the probability sequence $\{P^n\mu_0\}_{n\geq 1}$ will split into $d$ converging subsequences and has $d$ cyclic limiting probability distributions when $n \to \infty$, denoted as

$$\mu_\infty^r = \sum_{k=0}^{d-1} c_k (\xi^k)^r \vec{x}_k, \quad r = 0, 1, \ldots, d-1.$$

Thus, $P^n\mu_0$ is asymptotically periodic with period $d$ if $d > 1$ and asymptotically aperiodic if $d = 1$. □

We now consider a more general state space that may not necessarily be irreducible. According to the Decomposition theorem of the Markov chain [30], the finite state space $S$ can be partitioned uniquely as a set of transient states and one or several irreducible closed sets of recurrent states. According to [31], after performing an appropriate permutation of rows and columns, we can rewrite the transition probability matrix $P$ in its canonical form:

$$P = \left[\begin{array}{cccc|c} R_1 & \mathbf{0} & \cdots & \mathbf{0} & \mathbf{0} \\ \mathbf{0} & R_2 & \cdots & \mathbf{0} & \mathbf{0} \\ \vdots & \vdots & \ddots & \vdots & \vdots \\ \mathbf{0} & \mathbf{0} & \cdots & R_\alpha & \mathbf{0} \\ \hline T_1 & T_2 & \cdots & T_\alpha & Q \end{array}\right],$$

where $R_1, \ldots, R_\alpha$ represent the probability submatrices corresponding to the recurrent classes, Q represents the probability submatrix corresponding to the transient states, and $T_1, \ldots, T_\alpha$ represent the probability submatrices corresponding to the transitions between transient and recurrent classes $R_1, \ldots, R_\alpha$ respectively.

**Theorem 3.** *Suppose that the state space $\mathcal{S}$ is finite with a transition probability matrix $P \in \mathbb{R}^{|\mathcal{S}|\times|\mathcal{S}|}$ and $\mathcal{S}$ has $\alpha$ recurrent classes. Let $R_1, R_2, \ldots, R_\alpha$ be the probability submatrices corresponding to the recurrent classes and let $d_1, d_2, \ldots, d_\alpha$ be the number of the eigenvalues of modulus 1 that the submatrices $R_1, R_2, \ldots, R_\alpha$ has. Then for any initial distribution $\mu_0$, $P^n\mu_0$ is asymptotically periodic with period $d = \mathrm{lcm}(d_1, d_2, \ldots, d_\alpha)$ when $d > 1$ and asymptotically aperiodic when $d = 1$.*

*Proof.* Since $P$ is a block upper-triangular, it can be shown that the eigenvalues of $P$ are equal to the union of the eigenvalues of the diagonal blocks $R_1, \ldots, R_\alpha, Q$. Note that the $n$th-power of $P$ satisfies the following expression:

$$P^n = \left[\begin{array}{cccc|c} R_1^n & \mathbf{0} & \cdots & \mathbf{0} & \mathbf{0} \\ \mathbf{0} & R_2^n & \cdots & \mathbf{0} & \mathbf{0} \\ \vdots & \vdots & \ddots & \vdots & \vdots \\ \mathbf{0} & \mathbf{0} & \cdots & R_\alpha^n & \mathbf{0} \\ \hline T_1^{(n)} & T_2^{(n)} & \cdots & T_\alpha^{(n)} & Q^n \end{array}\right],$$

where $T_r^{(n)}$ is related to the $(n-1)$-th or the lower power of $R_r$ and $Q$. From Theorem 4.3 of [31], we obtain that $\lim_{n\to\infty} Q^n = \mathbf{0}$, which implies that all eigenvalues of $Q$ have modulus less than 1.

On the other hand, note that the sum of every row in matrix $R_r$ is equal to 1, which means $\lambda = 1$ is an eigenvalue of $R_r$ and all eigenvalues of $R_r$ satisfy $|\lambda| \leq 1$. Thus, the spectral radius of $P$ is equal to 1.

Note that the proof of Lemma 1 implies that the asymptotic periodicity of $P^n\mu_0$ depends on the eigenvalues of $P$ that have modulus 1. Since $R_r$ is non-negative irreducible with spectral radius 1, based on the Perron-Frobenius theorem used in Lemma 1, we can express the eigenvalues of $R_r$ in modulus 1 as:

$$\lambda_{r,k} = e^{j\frac{2\pi k}{d_r}}, \quad , k = 0, 1, \ldots, d_r - 1.$$

Based on the above discussion, it is easy to check that $\bigcup_{r=1}^{\alpha} \{\lambda_{r,0}, \ldots, \lambda_{r,d_r-1}\}$ is the set of all eigenvalues of modulus 1 of $P$. Rewrite $P$ in its Jordan canonical form $P = XJX^{-1}$, where

$$J = \begin{bmatrix} \lambda_{1,0} & & & & & & & & \\ & \ddots & & & & & & & \\ & & \lambda_{1,d_1-1} & & & & & & \\ & & & \lambda_{2,0} & & & & & \\ & & & & \ddots & & & & \\ & & & & & \lambda_{\alpha,d_\alpha-1} & & & \\ & & & & & & J_{d_1+\cdots+d_\alpha} & & \\ & & & & & & & \ddots & \\ & & & & & & & & J_s \end{bmatrix}$$

and $X = [\vec{x}_0, \vec{x}_1, \ldots, \vec{x}_{D-1}]$ is an invertible matrix. Similar to the proof in Lemma 1, we get

$$P^\infty \mu_0 = \lim_{n\to\infty} \sum_{r=1}^{\alpha} \sum_{k=0}^{d_r-1} c_k (e^{j\frac{2\pi k}{d_r}})^n \vec{x}_k := \lim_{n\to\infty} \mu^{(n)}.$$

Let $d = \mathrm{lcm}(d_1, d_2, \ldots, d_\alpha)$ and $r = n \pmod d$, then we have

$$\mu^{(n)} = \mu^{(r)}, \quad \forall n \geq 1.$$

Therefore, the probability sequence $\{P^n\mu_0\}_{n\geq 1}$ will split into $d$ converging subsequences and has $d$ cyclic limiting probability distributions when $n \to \infty$, denoted as

$$\mu_\infty^r = \sum_{r=1}^{\alpha} \sum_{k=0}^{d_r-1} c_k e^{j\frac{2\pi k r}{d_r}} \vec{x}_k, \quad r = 0, 1, \ldots, d-1.$$

Thus, $P^n\mu_0$ is asymptotically periodic with period $d$ if $d > 1$ and asymptotically aperiodic if $d = 1$. This completes the proof. □

### A.3 Proof of the Convergence of Our Auxiliary Loss

In this section, we provide a detailed derivation of the learning objective of SPF. As the DTFT of discrete-time state sequences is a continuous function that is difficult to compute, we practically sample the DTFT at $L$ equally-spaced points.

$$[\mathcal{F}\widetilde{s}_t]_k = \sum_{n=0}^{+\infty} [\widetilde{s}_t]_n \, e^{-j\frac{2\pi k}{L}n}, \quad k = 0, 1, \ldots, L-1. \tag{10}$$

As a result, the prediction target takes the form of a matrix with dimensions of $L * D$, where $D$ denotes the dimension of the state space. The auxiliary task is designed to encourage the representation to predict the Fourier transform of the state sequences using the current state-action pair as input. Specifically, we define the prediction target $F_{\pi,p}(s_t, a_t)$ as follows:

$$F_{\pi,p}(s_t, a_t) = \mathcal{F}\widetilde{s}(s_t, a_t) = \left\{ \sum_{n=0}^{+\infty} [\widetilde{s}(s_t, a_t)]_n \, e^{-j\frac{2\pi k}{L}n} \right\}_{k=0}^{L-1}, \tag{11}$$

For simplicity of notation, we substitute $F(s_t, a_t)$ for $F_{\pi,p}(s_t, a_t)$ in the following. We can derive that the DTFT functions at successive time steps are related to each other in a recursive form:

$$\begin{aligned}
[F(s_t, a_t)]_k &= \sum_{n=0}^{+\infty} \gamma^n \cdot e^{-j\frac{2\pi k}{L}n} \cdot E_{\pi,p}\left[s_{t+n+1} \middle| s_t = s, a_t = a\right] \\
&= E_p\left[s_{t+1} \middle| s_t = s, a_t = a\right] + \gamma \cdot e^{-j\frac{2\pi k}{L}} \cdot \\
&\quad E_{s_{t+1}\sim p, a_{t+1}\sim\pi}\left[\sum_{n=0}^{+\infty} \gamma^n \cdot e^{-j\frac{2\pi k}{L}n} \cdot E_p\left[s_{t+n+2} \middle| s_{t+1}, a_{t+1}\right]\right] \\
&= [\widetilde{s}_t]_0 + \gamma \cdot e^{-j\frac{2\pi k}{L}} \cdot E_{\pi,p}\left[[F(s_{t+1}, a_{t+1})]_k\right], \ \forall\, k = 0, 1, \ldots L-1.
\end{aligned}$$

We can further express the above equation as a matrix-form recursive formula as follows:

$$F(s_t, a_t) = \widetilde{\boldsymbol{S}}_t + \Gamma \, E_{\pi,p}\left[F(s_{t+1}, a_{t+1})\right], \tag{12}$$

where

$$\widetilde{\boldsymbol{S}}_t = [[\widetilde{s}_t]_0, \ldots, [\widetilde{s}_t]_0]^\mathsf{T} \in \mathbb{R}^{L\times D},$$

$$\Gamma = \gamma \begin{bmatrix} 1 & & & & \\ & e^{-j\frac{2\pi}{L}} & & & \\ & & e^{-j\frac{4\pi}{L}} & & \\ & & & \ddots & \\ & & & & e^{-j\frac{(L-1)\pi}{L}} \end{bmatrix}.$$

Similar to the TD-learning of value functions, we can prove that the above recursive relationship (12) can be reformulated as a contraction mapping $\mathcal{T}$. Due to the properties of contraction mappings, we can iteratively apply the operator $\mathcal{T}$ to compute the target DTFT function until convergence in tabular settings.

**Theorem 4.** *Let $\mathcal{F}$ denote the set of all functions $F : \mathcal{S} \times \mathcal{A} \to \mathbb{C}^{L*D}$ and define the norm on $\mathcal{F}$ as*

$$\|F\|_{\mathcal{F}} := \sup_{\substack{s\in\mathcal{S} \\ a\in\mathcal{A}}} \max_{0\le k<L} \left\| \left[F(s, a)\right]_k \right\|_D,$$

*where $\left[F(s, a)\right]_k$ represents the kth row vector of $F(s, a)$. We show that the mapping $\mathcal{T} : \mathcal{F} \to \mathcal{F}$ defined as*

$$\mathcal{T}F(s_t, a_t) = \widetilde{\boldsymbol{S}}_t + \Gamma \, E_{\pi,P}\left[F(s_{t+1}, a_{t+1})\right] \tag{13}$$

*is a contraction mapping, where $\widetilde{\boldsymbol{S}}_t$ and $\Gamma$ are defined as above.*

*Proof.* For any $F_1, F_2 \in \mathcal{F}$, we have

$$\begin{aligned}
\|\mathcal{T}F_1 - \mathcal{T}F_2\|_{\mathcal{F}} &= \sup_{\substack{s\in\mathcal{S}\\a\in\mathcal{A}}} \max_{0\le k<L} \Bigg\| s + \gamma e^{-j\frac{2\pi k}{L}} E_{\substack{s'\sim P(\cdot|s,a)\\a'\sim\pi(\cdot|s')}} \left[ \left[F_1(s',a')\right]_k \Big| s,a \right] \\
&\qquad - s - \gamma e^{-j\frac{2\pi k}{L}} E_{\substack{s'\sim P(\cdot|s,a)\\a'\sim\pi(\cdot|s')}} \left[ \left[F_2(s',a')\right]_k \Big| s,a \right] \Bigg\|_D \\
&\le \gamma \cdot \max_{0\le k<L} \sup_{\substack{s\in\mathcal{S}\\a\in\mathcal{A}}} \Bigg\| E_{\substack{s'\sim P(\cdot|s,a)\\a'\sim\pi(\cdot|s')}} \left[ \left[F_1(s',a')\right]_k - \left[F_2(s',a')\right]_k \Big| s,a \right] \Bigg\|_D \\
&\le \gamma \cdot \max_{0\le k<L} \sup_{\substack{s'\in\mathcal{S}\\a'\in\mathcal{A}}} \left\| \left[F_1(s',a') - F_2(s',a')\right]_k \right\|_D \\
&= \gamma \cdot \|F_1 - F_2\|_{\mathcal{F}}.
\end{aligned}$$

Note that $\gamma \in [0, 1)$, which implies that $\mathcal{T}$ is a contraction mapping. □

# B Pseudo-code of SPF

The training procedure of SPF is shown in the pseudo-code as follows:

**Algorithm 1** State Sequences Prediction via Fourier Transform (SPF)

Denote parameters of the online encoder $(\phi_s, \phi_{s,a})$, predictor $\mathcal{F}$, and projection $\psi$ as $\theta_{\text{aux}}$
Denote parameters of the target encoder $(\widehat{\phi}_s, \widehat{\phi}_{s,a})$, predictor $\widehat{\mathcal{F}}$, and projection $\widehat{\psi}$ as $\widehat{\theta}_{\text{aux}}$
Denote parameters of actor model $\pi$ and critic model $Q$ for RL agents as $\theta_{\text{RL}}$
Denote the smoothing coefficient and update interval for target network updates as $\tau$ and $K$
Initialize replay buffer $\mathcal{D}$ and parameters $\theta_{\text{aux}}, \theta_{\text{RL}}$
**for** each environment step $t$ **do**
  $a_t \sim \pi(\cdot|\phi_s(s_t))$
  $s_{t+1}, r_{t+1} \sim p(\cdot|s_t, a_t)$
  $\mathcal{D} \leftarrow \mathcal{D} \cup (s_t, a_t, s_{t+1}, r_{t+1})$
  sample a minibatch of $\{(s_t, a_t, s_{t+1}, r_{t+1})\}$ from $\mathcal{D}$
  $\theta_{\text{aux}} \leftarrow \theta_{\text{aux}} - \alpha_{\text{aux}} \nabla_{\theta_{\text{aux}}} L_{\text{pred}}(\theta_{\text{aux}}, \widehat{\theta}_{\text{aux}})$
  resampling a minibatch of $\{(s_t, a_t, s_{t+1}, r_{t+1})\}$ from $\mathcal{D}$
  $\overline{s_t} \leftarrow \phi_s(s_t)$
  $z_{s_t,a_t} \leftarrow \phi_{s,a}(\phi_s(s_t), a_t)$
  update the RL agent parameters $\theta_{\text{RL}}$ with the representations $\overline{s_t}$, $z_{s_t,a_t}$
  update parameters of target networks with $\widehat{\theta}_{\text{aux}} \leftarrow \tau\theta_{\text{aux}} + (1-\tau)\widehat{\theta}_{\text{aux}}$ every $K$ steps
**end for**

# C Network Details

The encoders $\phi_s$ and $\phi_{s,a}$ share the same architecture. Each layer of the encoders uses MLP-DenseNet [19], a slightly modified version of DenseNet. For each MuJoCo task, the incremental number of hidden units per layer is selected from $\{30, 40\}$, while the number of layers is selected from $\{6, 8\}$ (see Table 2). Both the predictor $\mathcal{F}$ and the projection $\psi$ apply a 2-layer MLP. We divide the last layer of the predictor into two heads as the real part $\mathcal{F}_{\text{Re}}$ and the imaginary part $\mathcal{F}_{\text{Im}}$, respectively, since the prediction target of our auxiliary task is complex-valued. With respect to the projection module, we add an additional 2-layer MLP (referred to as *Projection2*) after the original online projection to perform a dimension-invariant nonlinear transformation on the predicted DTFT that has been projected to a lower-dimensional space. We do not apply this nonlinear operation to the target projection. This additional step is carried out to prevent the projection from collapsing to a constant value in the case where the online and target projections share the same architecture.

In Fourier analysis, the low-frequency components of the DTFT contain information about the long-term trends of the signal, with higher signal energy, while the high-frequency components of the

Table 2: Detailed setting of the encoder for six MuJoCo tasks.

| Environment | Number of Layers | Number of Units per Layer | Activation Function |
|---|---|---|---|
| HalfCheetah-v2 | 8 | 30 | Swish |
| Walker2d-v2 | 6 | 40 | Swish |
| Hopper-v2 | 6 | 40 | Swish |
| Ant-v2 | 6 | 40 | Swish |
| Swimmer-v2 | 6 | 40 | Swish |
| Humanoid-v2 | 8 | 40 | Swish |

DTFT reflect the amount of short-term variation present in the state sequences. Therefore, we attempt to preserve the overall information of the low and high-frequency components of the predicted DTFT by directly computing the cosine similarity distance without undergoing the dimensionality reduction process. For the remaining frequency components of the predicted DTFT, we first utilize projection layers to perform dimensionality reduction, followed by calculating the cosine similarity distance. The sum of these three distances is used as the final loss function, which we call *freqloss*.

# D Hyperparameters

Table 3: Hyperparameters of auxiliary prediction tasks.

| Hyperparameter | Setting |
|---|---|
| Optimizer | Adam |
| Discount $\gamma$ | 0.99 |
| Learning rate | 0.0003 |
| Number of batch size | 256 |
| Predictor: Number of hidden layers | 1 |
| Predictor: Number of hidden units per layer | 1024 |
| Predictor: Activation function | ReLU |
| Projection: Number of hidden layers | 1 |
| Projection: Number of hidden units per layer | 512 |
| Projection: Activation function | ReLU |
| Projection2: Number of hidden layers | 1 |
| Projection2: Number of hidden units per layer | 512 |
| Projection2: Activation function | ReLU |
| Number of discrete points for sampling the DTFT $L$ | 128 |
| The dimensionality of the output of projection | 512 |
| Replay buffer size | 100,000 |
| Pre-training steps | 10000 |
| Target smoothing coefficient $\tau$ | 0.01 |
| Target update interval $K$ | 1000 |
| *Hyperparameters of SPF-SAC* | |
| Each module: Normalization Layer | BatchNormalization |
| Random collection steps before pre-training | 10,000 |
| *Hyperparameters of SPF-PPO* | |
| Each module: Normalization Layer | LayerNormalization |
| Random collection steps before pre-training | 4,000 |
| $\theta_{\text{aux}}$ update interval $K_2$ | |
| HalfCheetah-v2 | 5 |
| Walker2d-v2 | 2 |
| Hopper-v2 | 150 |
| Ant-v2 | 150 |
| Swimmer-v2 | 200 |
| Humanoid-v2 | 1 |

We select $L = 128$ as the number of discrete points sampled over one period of DTFT. In practice, due to the symmetry conjugate of DTFT, the predictor $\mathcal{F}$ only predicts $\frac{L}{2} + 1$ points on the left half of our frequency map, as mentioned in Section 5.2. The projection module described in Section 5.3 projects the predicted value, a matrix with the dimension of $L * D$, into a 512-dimensional vector. To update target networks, we overwrite the target network parameters with an exponential moving average of the online network parameters, with a smoothing coefficient of $\tau = 0.01$ for every $K = 1000$ steps.

In order to eliminate dependency on the initial parameters of the policy, we use a random policy to collect transitions into the replay buffer [32] for the first 10K time steps for SAC, and 4K time steps for PPO. We also pretrain the representations with the aforementioned random collected samples to stabilize inputs to each RL algorithm, as described in [19].

The network architectures, optimizers, and hyperparameters of SAC and PPO are the same as those used in their original papers, except that we use mini-batches of size 256 instead of 100. As for PPO, we perform $K_2$ gradient updates of $\theta_{\text{aux}}$ for every $K_2$ steps of data sampling. The update interval $K_2$ is set differently for six MuJoCo tasks and can be found in Table 3.

# E Visualization

To demonstrate that the representations learned by SPF effectively capture the structural information contained in infinite-step state sequences, we compare the true state sequences with the states recovered from the predicted DTFT via the inverse DTFT.

Specifically, we first generate a state sequence from the trained policy and select a goal state $s_t$ at a certain time step. Next, we choose a historical state $s_{t-k}$ located k steps past the goal state and select an action $a_{t-k}$ based on the trained policy $\pi(\cdot|s_{t-k})$ as the inputs of our trained predictor. We then obtain the DTFT $F_{t-k} := F_\pi(s_{t-k}, a_{t-k})$ of state sequences starting from the state $s_{t-k+1}$. Next, we compute the kth element of the inverse DTFT of $F_{t-k}$ and obtain a recovered state $\widehat{s}_t$, which represents that we predict the future goal state using the historical state located k steps past the goal state. By selecting a sequence of states over a specific time interval as the goal states and repeating the aforementioned procedures, we will obtain a state sequence recovered by k-step prediction. In Figure 5(b), 6(b), 7(b), 8(b), 9(b) and 10(b), we visualize the true state sequence (the blue line) and the recovered state sequences (the red lines) via k-step predictions for $k = 1, 2, 3, 4, 5$. Note that the lighter red line corresponds to predictions made by historical states from a more distant time step. We conduct the visualization experiment on six MuJoCo tasks using the representations and predictors trained by SPF-SAC or SPF-PPO. Due to the large dimensionality of the states in Ant-v2 and Humanoid-v2, which contain many zero values, we have chosen to visualize only six dimensions of their states, respectively. The fine distinctions between the true state sequences and the recovered state sequences from our trained representations and predicted FT indicates that our representation effectively captures the inherent structures of future state sequences.

Furthermore, we provide a visualization that compares the true DTFT and the predicted DTFT in Figure 5(a), 6(a), 7(a), 8(a), 9(a) and 10(a). To accomplish this, we use our trained policies to interact with the environments and select the state sequences of the 200 last steps of an episode. The blue lines represent the true DTFT of these state sequences, while the orange line represents the predicted DTFT using the online encoder and predictor trained by our learned policies. It is evident that the true DTFT and the predicted DTFT exhibit significant differences. These results demonstrate the ability of SPF to effectively extract the underlying structural information in infinite-step state sequences without relying on high prediction accuracy.

# F Cosine Similarity Results

We present a comparison table between the real discrete-time Fourier transform (DTFT) and the predicted DTFT. To obtain these results, we use our trained policies to interact with six MuJoCo environments and record the states and the actions for the 201 last steps of an episode, denoted as $\{s_t\}_{t=0}^{200}$ and $\{a_t\}_{t=0}^{200}$, respectively. The real DTFT of the state sequence $\{s_1, s_2, \ldots, s_{200}\}$ over a 200-step horizon is computed using the formulation of the discrete-time Fourier transform. For the predicted DTFT, we employ our trained encoder and prediction module, with the state $s_0$ and the action $a_0$ serving as inputs. The output of the prediction module represents the predicted DTFT of the state sequence $\{s_1, s_2, \ldots, s_{200}\}$. We compute the cosine similarity between $A$ and $B$ with the

formula $-\frac{A \cdot B}{\|A\|\|B\|}$, where values closer to $-1$ indicate greater similarity while the values closer to 1 indicate greater dissimilarity. The cosine similarity between the real DTFT and the predicted DTFT is listed in Table 4 below.

Table 4: Cosine similarity distance between the real DTFT and the predicted DTFT.

| Environment | Cosine similarity distance |
|---|---|
| HalfCheetah | $-0.655$ |
| Hopper | $-0.660$ |
| Walker2d | $-0.651$ |
| Ant | $-0.351$ |
| Swimmer | $-0.557$ |
| Humanoid | $-0.651$ |

# G Code

Codes for the proposed method are available at https://github.com/MIRALab-USTC/RL-SPF/.

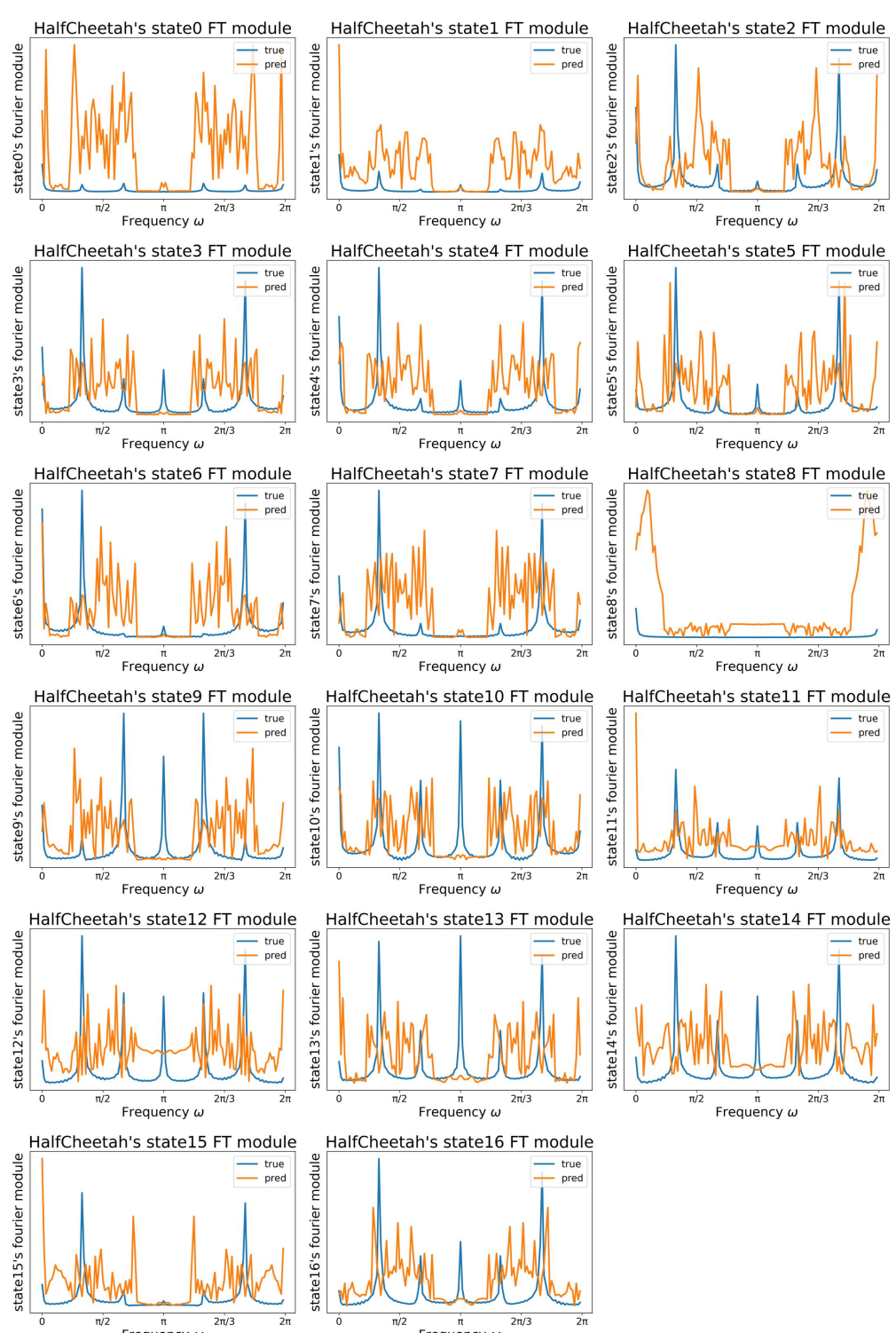

(a) True and predicted DTFT

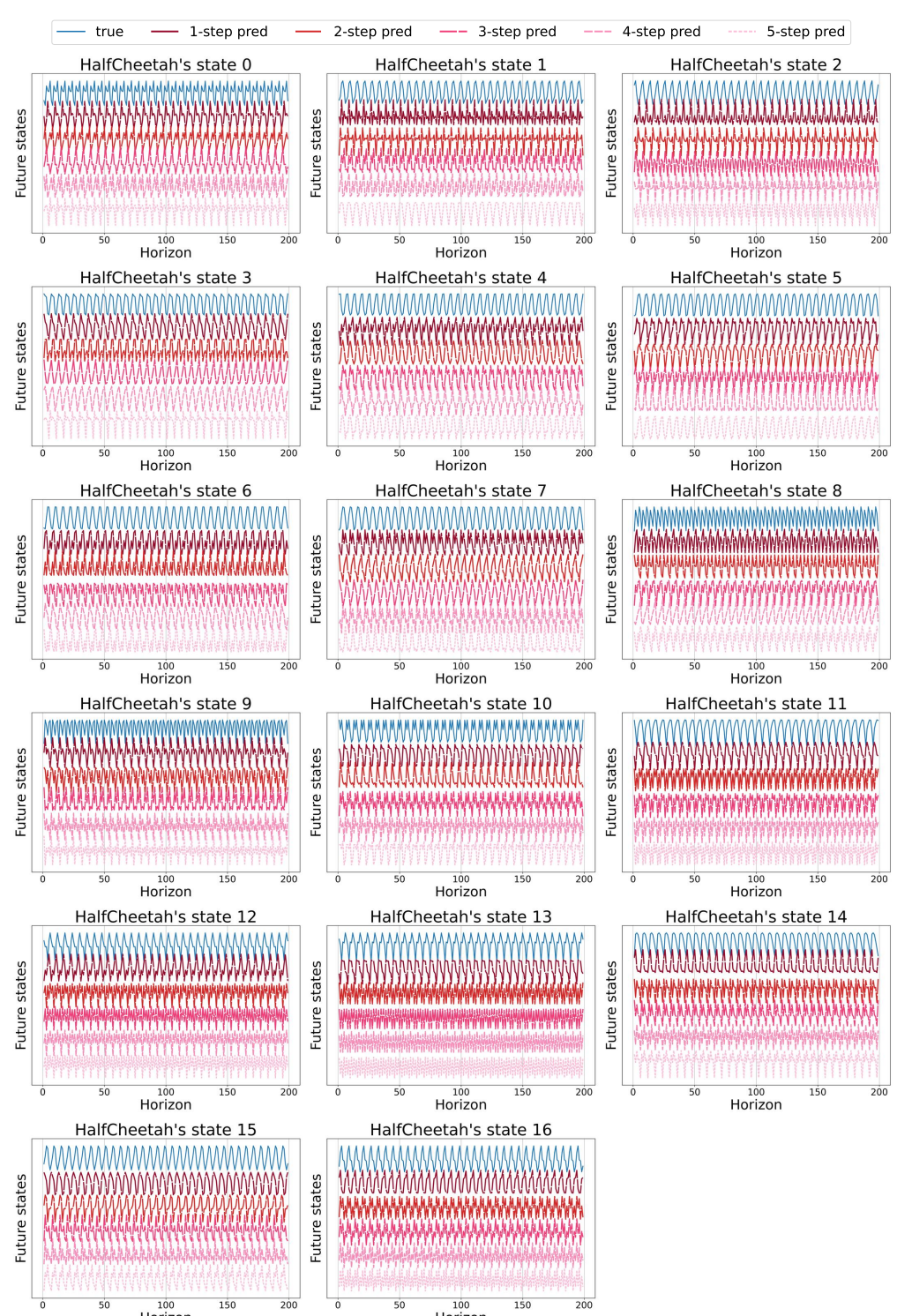

(b) True and recovered state sequences

Figure 5: Predicted values via representations trained by SPF-SAC on HalfCheetah-v2

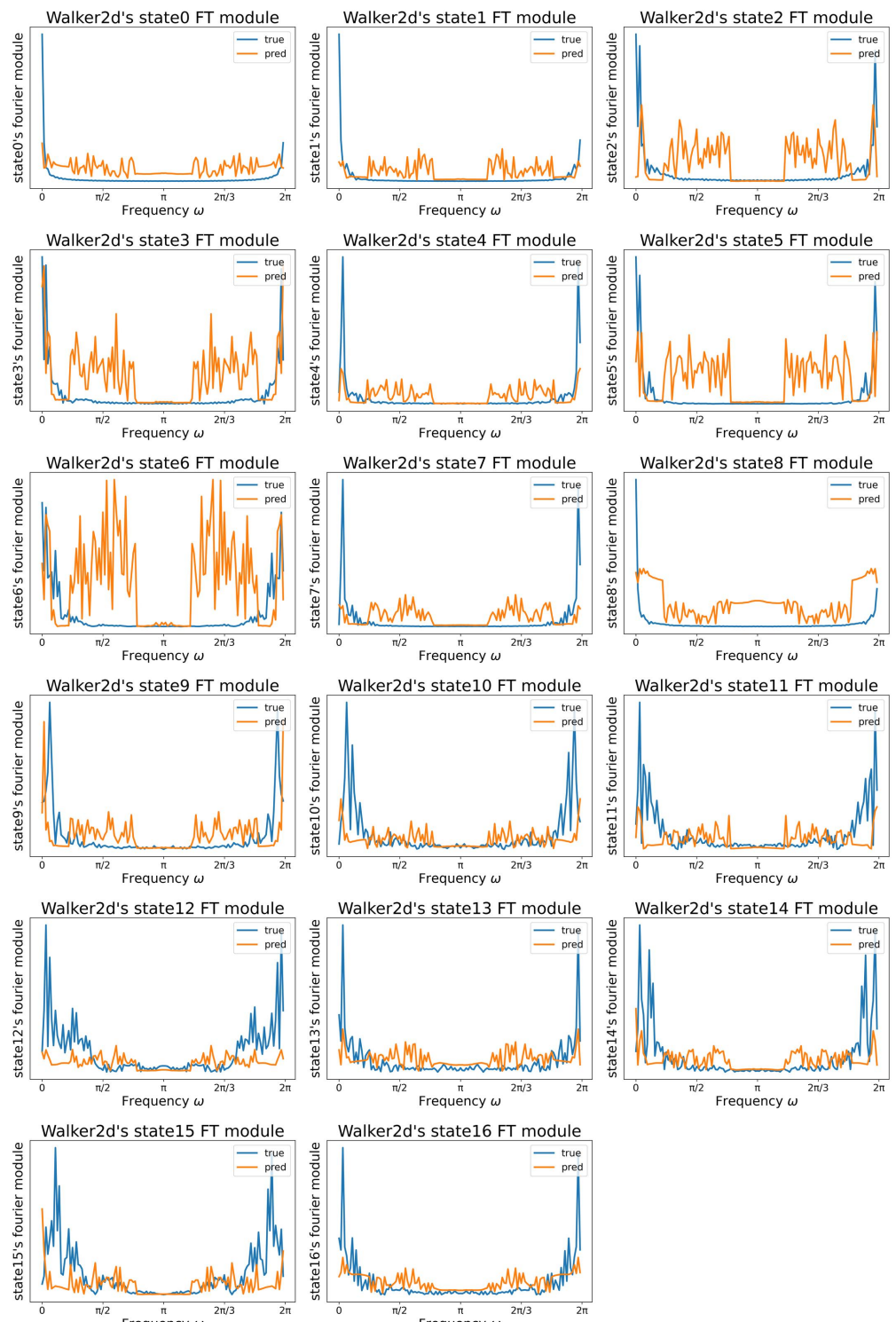

(a) True and predicted DTFT

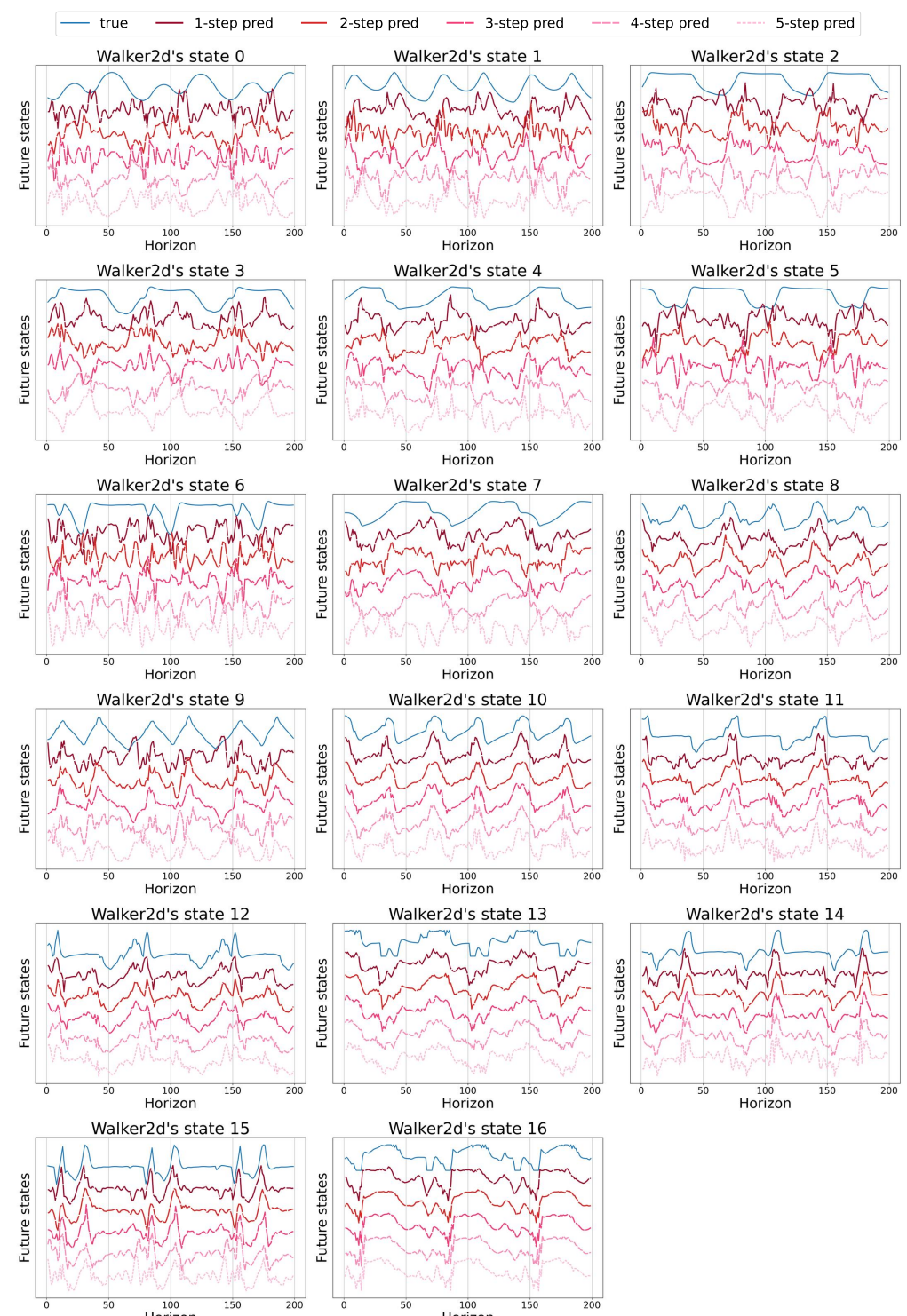

(b) True and recovered state sequences

Figure 6: Predicted values via representations trained by SPF-SAC on Walker2d-v2

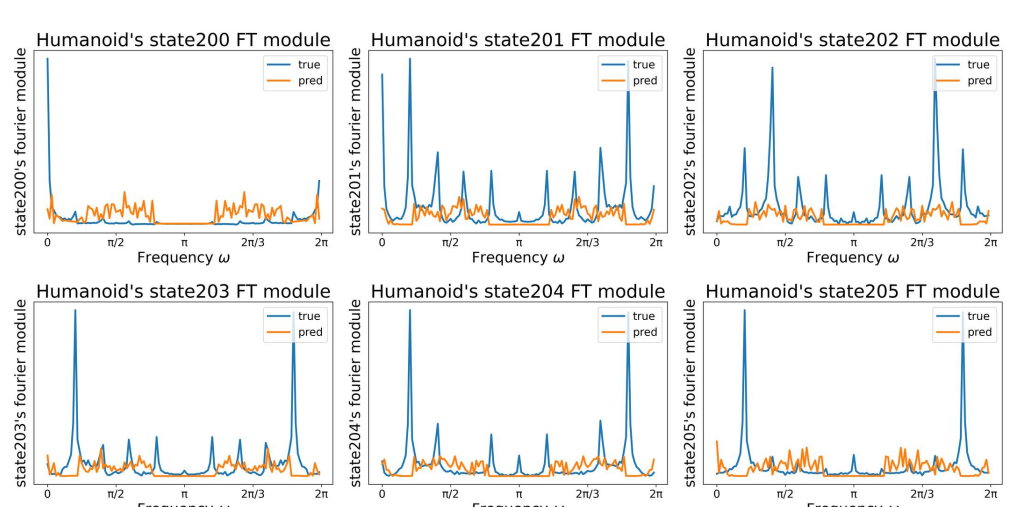

(a) True and predicted DTFT

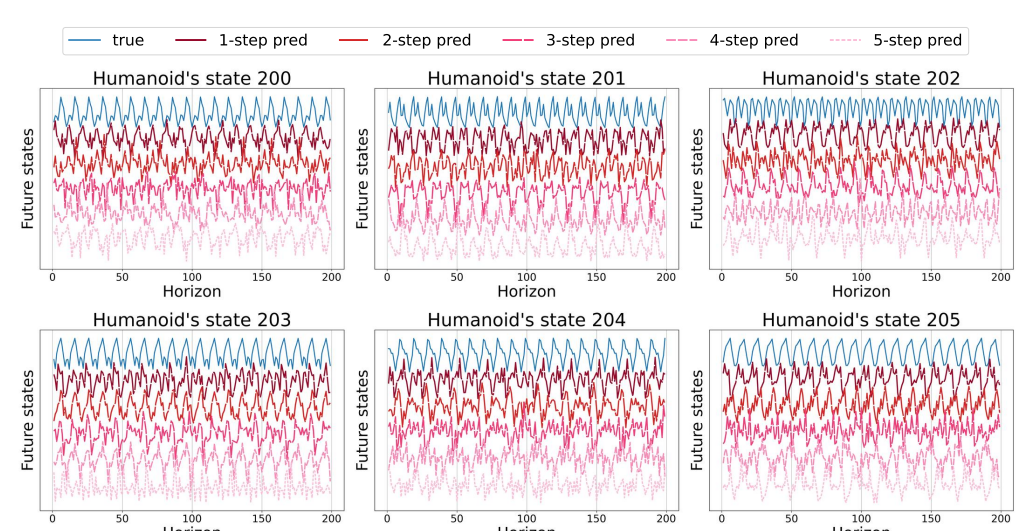

(b) True and recovered state sequences

Figure 7: Predicted values via representations trained by SPF-SAC on Humanoid-v2

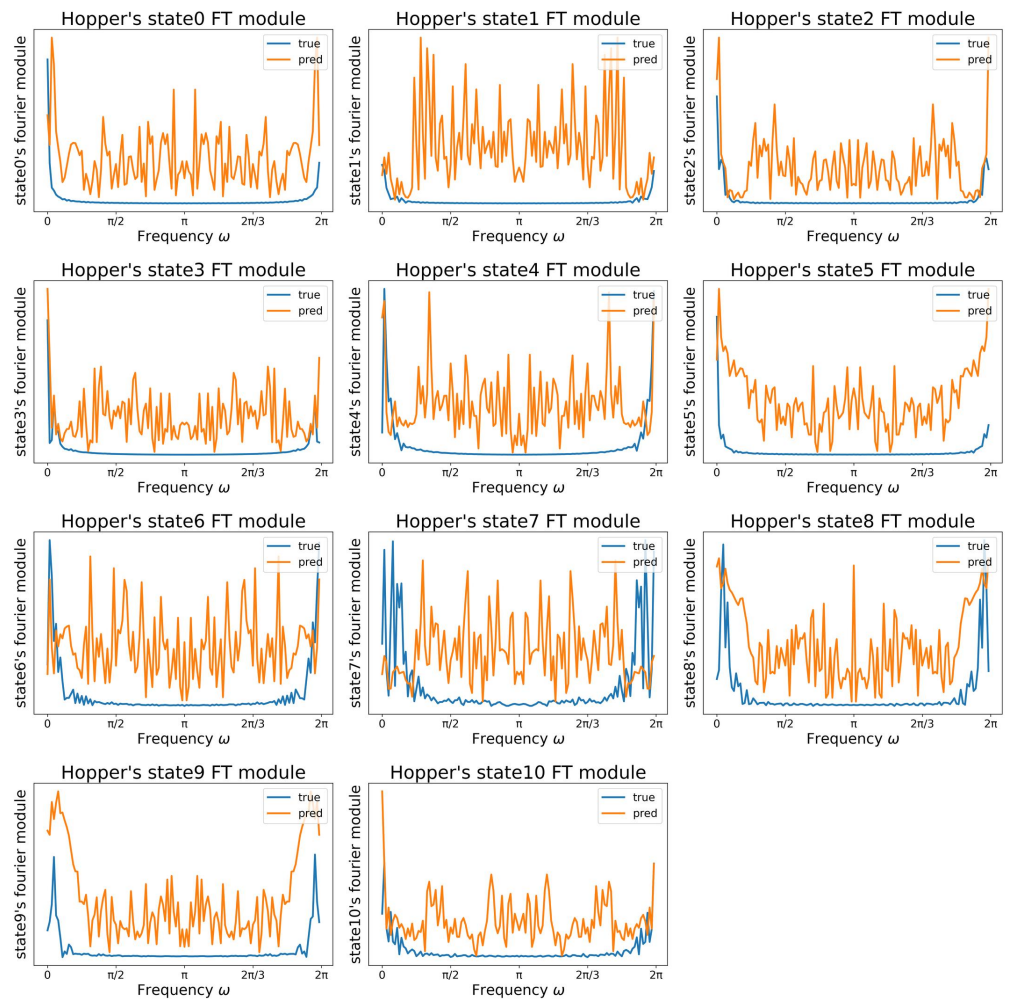

(a) True and predicted DTFT

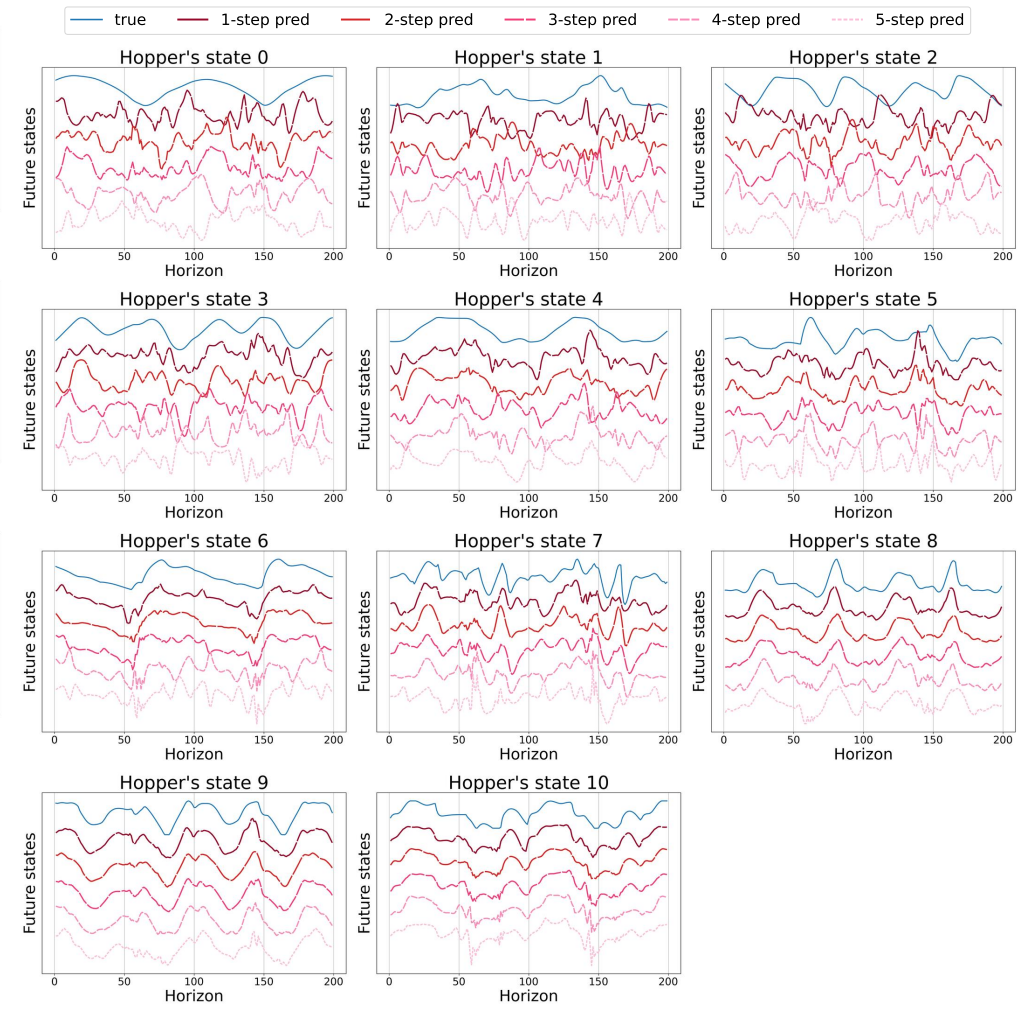

(b) True and recovered state sequences

Figure 8: Predicted values via representations trained by SPF-PPO on Hopper-v2

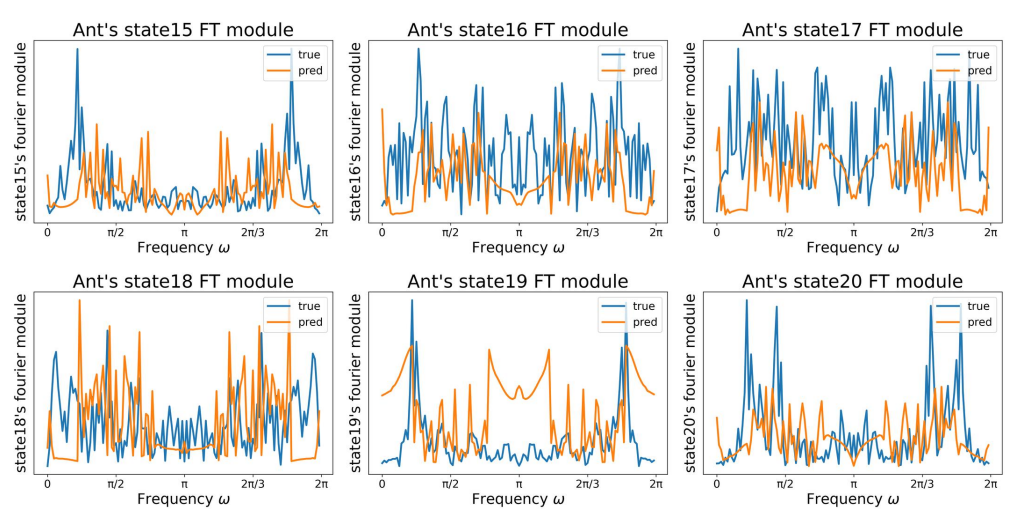

(a) True and predicted DTFT

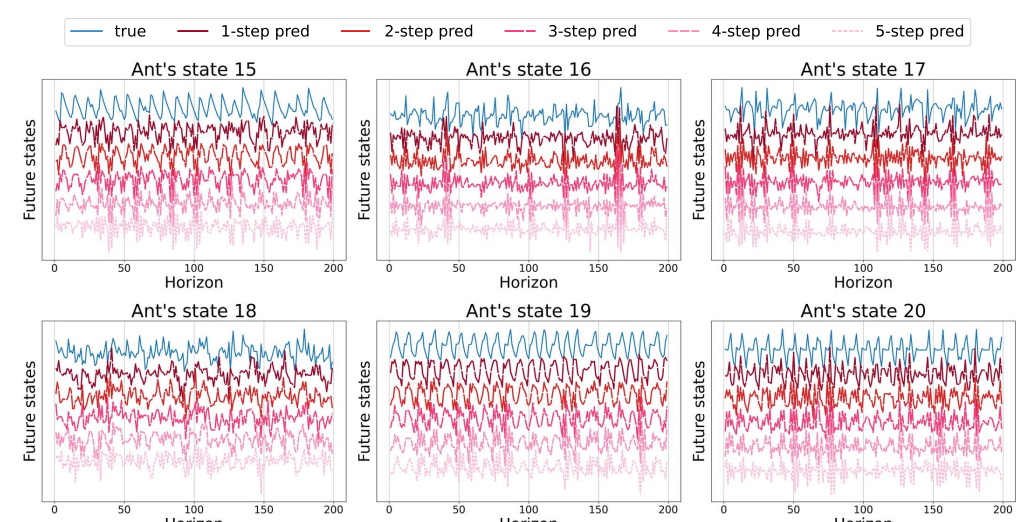

(b) True and recovered state sequences

Figure 9: Predicted values via representations trained by SPF-PPO on Ant-v2

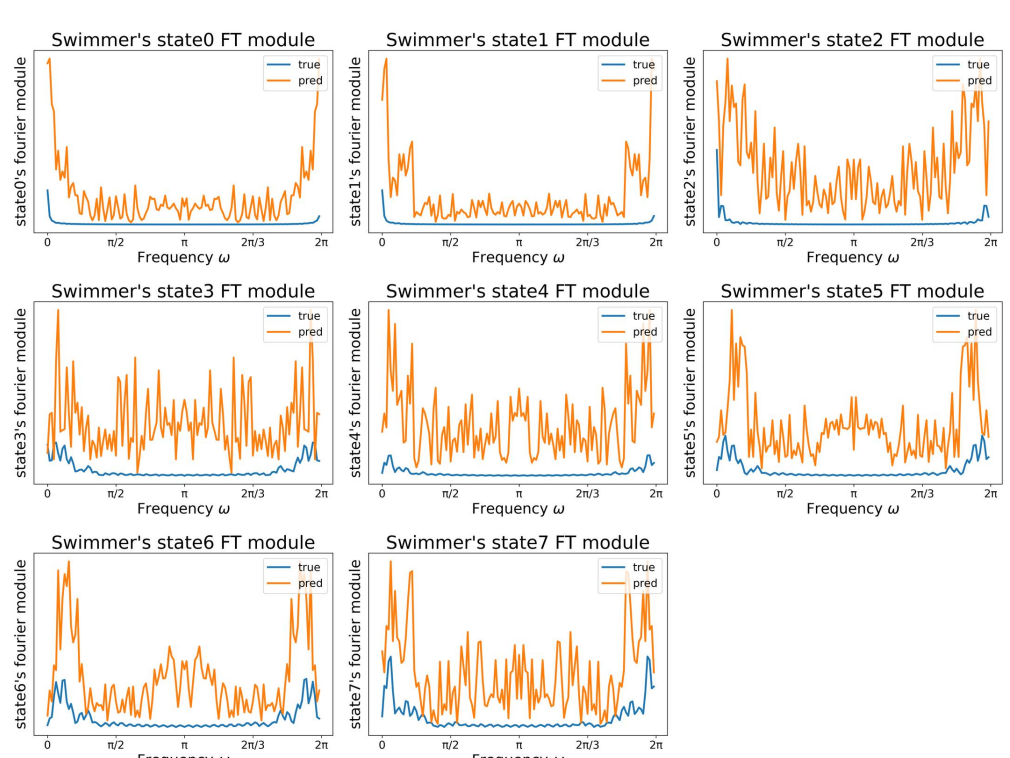

(a) True and predicted DTFT

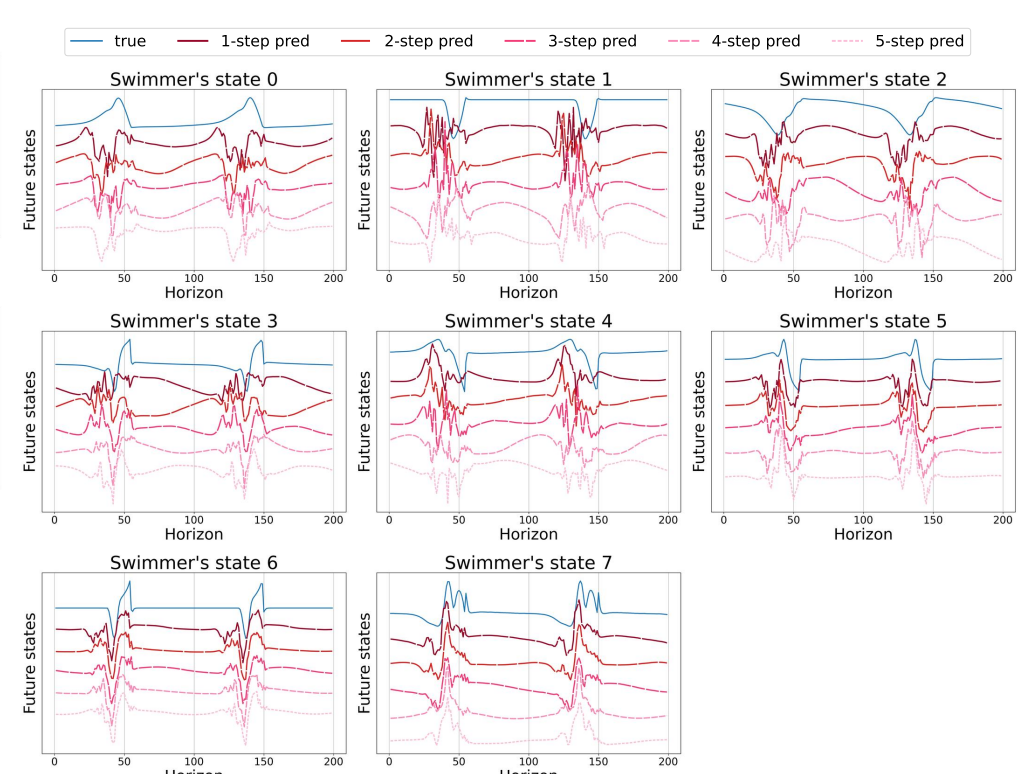

(b) True and recovered state sequences

Figure 10: Predicted values via representations trained by SPF-PPO on Swimmer-v2