# OpenReview forum: "State Sequences Prediction via Fourier Transform for Representation Learning"
_NeurIPS.cc/2023/Conference — NeurIPS 2023 spotlight_

### Official Review · Reviewer_i4e2 · 2023-07-01

**Soundness:** 2 fair
**Presentation:** 2 fair
**Contribution:** 2 fair
**Rating:** 5
**Confidence:** 4

**Summary:**

The authors in this paper investigated the structural information in state sequences for reinforcement learning, and proposed the prediction of Fourier transformation for the input sequences, which is further used an auxiliary task for learning policies. Some theoritical results are provided to show the existence of structual information. Experimental results on MuJoKo tasks demonstrated the promise of the proposed method.


**Strengths:**

1. The idea of performing prediction on the Fourier spectrum is interesting. This approach may better extract information hidden in the time domain, which constitutes the main contribution of this paper.

2. From the experimental results, the proposed auxiliary task indeed helped learn better policies for RL agents, w.r.t. both SAC and PPO.

**Weaknesses:**

1. There seems to be a gap between the motivation and the implementation. The introduction and some theoretical results, e.g., Theorem 3, imply that the prediciton is on the Fourier domain; however, the algorithm (SPF) simply used a parameterized prediction module. Consequently, it's unclear to me whether the proposed algorithm can extract the information in the Fourier spectrum.

2. It's unclear to me whether the proposed method can capture the long-term information. From Figure 2, it only considers two adjacent states as inputs.

3. The significance in Theorem 1 may not be sufficient, as the upper bound can be loose. Note that the use of l1 norm for the difference between two distributions will make the term in Equation (3) much greater than 1. Furthermore, it's unclear whether the states distributions for two policies can vary that much, given the use of Rmax / (1 - \gamma).

**Questions:**

1. Some notations in Figure 2 are not defined (e.g., $P$) and cannot be located from the definition of the prediction loss (e.g., $\mathcal{F}$) between L261 and L262.

2. What does $\Gamma_{Re}$ do in the prediction loss, given that $\mathcal{F}_{Re}$ is already real?

**Limitations:**

Yes, the authors mentioned that the current version may not handle the image inputs.

---

> ### Author Rebuttal · Authors · 2023-08-10
>
> We thank the reviewer for the positive and insightful comments. We respond to each comment as follows and sincerely hope that our rebuttal could properly address your concerns. If so, we would deeply appreciate it if you could raise your score. If not, please let us know your further concerns, and we will continue actively responding to your comments and improving our submission.
>
> **Q1. The gap between the motivation and the implementation.**
>
> **A1.**
> + To narrow the gap between the motivation and the implementation, we provide **a sub-optimality performance bound for the latent policy using the learned Fourier representation** as follows:
>
>   *Suppose that $\mathcal{S}\subset\mathbb{R}^D$ the reward function $R(s,a,s')=R(s)$ is a linear function with respect to $s\in \mathcal{S}$. Then the performance difference between the optimal policy* $\pi^*$ *and the latent policy* $\overline{\pi}\circ \phi $ *may be bounded as:*
>   $$
>   |J(\pi^*)-J(\overline{\pi}\circ\Phi)|\leq \frac{\sqrt{D}\cdot R_\text{max}}{(1-\gamma)^2}\cdot E_{(s,a,s')\sim\mathcal{D}}\left[\overline{\delta}_{\overline{\pi}\circ\phi }(s,a,s')\right],
>   $$
>
>   *where $\overline{\delta}_\pi(s,a,s'):=s'+e^{-j\omega}\gamma E\_{a'\sim{\pi}}\left[F\_{\pi}(s',a')\right] - F\_{\pi}(s,a)$ is the minimized objective of SPF.*
>
>   **We provide detailed proof in the 'global' response.**
>
> + Based on the above theorem, the performance difference between the optimal policy and our learned policy using Fourier representation decreases when we minimize the representation learning objective of SPF (Line 261 in our paper). This indicates that leveraging Fourier spectrum information for representation learning aids the agent in learning the optimal policy.
>
> + In practice, we use the predictor module $\mathcal{F}$ to estimate the Fourier Transform (FT) of the state sequence $F_{\pi}$ with the latent state $\phi(s)$ and action $a$ as input. Thus, we rewrite the objective as
>
>   $$
>   \overline{\delta}_\pi(s,a,s'):=s'+e^{-j\omega}\gamma E\_{a'\sim{\pi}}\left[\mathcal{F}(\phi(s'),a')\right] - \mathcal{F}(\phi(s),a).
>   $$
>
> + Note that we use the objective $\overline{\delta}_\pi(s,a,s')$ to **update the encoder $\phi$ and the predictor $\mathcal{F}$ simultaneously**. Thus, the information in the Fourier spectrum of state sequences is propagated to the encoder when minimizing the objective of SPF.
>
>
>
> **Q2. Explanation for Figure 2.**
>
> **A2.**
> + Thank you for pointing out this typo. We apologize for any inconvenience or confusion this may have caused. In Figure 2 of the main text, the letter $\mathcal{P}$ (including $\mathcal{P}\_\text{Re}$ and $ \mathcal{P}\_{\text{Im}}$) should be $\mathcal{F}$ (also $\mathcal{F}\_{\text{Re}}$ and $\mathcal{F}\_{\text{Im}}$), which denotes the predictor.
>
> + **The prediction loss in Figure 2 corresponds to the objective $\overline{\delta}$ mentioned in A1 of this response.** It is derived from the recursive form of the FT of state sequences (See Equation (6)), and it can be expressed as the difference between the estimated Fourier value of $F_{\pi}(s,a)$ and the better estimate $s'+e^{-j\omega}\gamma E_{a'\sim{\pi}}\left[F_{\pi}(s',a')\right]$, just like the TD error. **When calculating the prediction loss, we only need to utilize the current state, the current action, and the next state**. Thus, one of the advantages of SPF is without the need to store multi-step future states as labels for predicting future state sequences.
>
> + Similar to how the classic RL algorithms use TD error to update the $Q(s,a)$ function for estimating the sum of future rewards, **our prediction loss enables the predictor to estimate the FT of long-term state sequences from the tuple $(s,a,s')$.** Theorem 4 in our paper shows that the recursive operator $\mathcal{T}$ is a contraction mapping, which guarantees convergence to the true FT function in tabular settings.
>
>
>
> **Q3. The upper bound of Theorem 1.**
>
> **A3.**
> + **Theorem 1 provides intuition about the relationship between the distribution of the state sequences and policy performance.** As the state sequences generated by the learned policy approach the state sequences generated by the optimal policy, the performance difference between the two policies tends to zero. However, in online RL, we have no access to the state sequences generated by the optimal policy. Thus, **in practice, we optimize $\overline{\delta}_\pi$ (the TD-style loss) instead of the right term of Theorem 1.** In A1 of this response, we present a theorem that establishes an upper bound on the sub-optimality performance gap using our optimized objective.
>
> + According to  Lemma 13.3 in [1], the L1 norm is equivalent to the total variation (TV) norm. The total variation norm is a widely used distance measure for probability distributions.
>
> + From the proof of Theorem 1, we acknowledge that the constant $\frac{R_\text{max}}{1-\gamma}$ is necessary to establish the bound. In practice, we can consider constraining the reward range to minimize the magnitude of this constant.
>
>
>
> **Q4. What does $\Gamma_\text{Re}$ do in the prediction loss?**
>
> **A4.**
> Considering that the states are vectors and we only take discrete samples for the estimation of FT, we rewrite the prediction loss in the matrix form $\overline{\delta}_\pi(s_t,a_t,s_t') = \widetilde{\boldsymbol{S}}\_t + \Gamma \, E\_{\pi,p}\left[F(s\_{t+1},a\_{t+1}) \right] - F(s_t,a_t)$. The form of  $\Gamma$ is on Line 565 of the Appendix.
>
> Note that the prediction loss $\overline{\delta}\_\pi$ is complex-valued as $F_\pi$ and $\Gamma$ are complex-valued matrices. Thus, we measure the magnitude of $\overline{\delta}\_\pi$ with its modulus, that is, the sum of the squares of its real and imaginary parts. So we have to express $\Gamma=\Gamma\_{\text{Re}}+i\Gamma\_{\text{Re}}$ and  $F=F\_{\text{Re}}+i F\_{\text{Re}}$ to compute the modules of the prediction loss.
>
>
>
> [1] Driver B K. Math 280 (Probability Theory) Lecture Notes. Lecture notes, Department of Mathematics.

---

> > ### Comment · Reviewer_i4e2 · 2023-08-19
> >
> > Thanks to the authors for the response, I really appreciate it.
> >
> > Regarding the gap between motivation and implementation, it is still unclear to me how well the module $\mathcal{F}$ can approximate the real Fourier transformation. For example, how about their distance between prediction and ground truth?
> >
> > For the upper bound in Theorem 1, my point is that the existence of $\frac{R_max}{1 - \gamma}$ just weakens the claim, as it could dominate the performance gap between $\pi_1$ and $\pi_2$, which makes the last term less relevant.

---

> > > ### Author Response · Authors · 2023-08-20
> > > **Thanks for your comments.**
> > >
> > > Thank you for your insightful comments and for helping us gain deeper insights into our work. We respond to your comments as follows and sincerely hope that our response could properly address your concerns.
> > >
> > > **Q1. How well the module $\mathcal{F}$ can approximate the real Fourier transformation?**
> > >
> > > **A1.**
> > >
> > > + Given that 1) the parameters of the policy $\pi$ are updated alternately during the training of the prediction module $\mathcal{F}$; 2) the policy $\pi$ is trained to approach the optimal policy $\pi^*$. The ground truth for our prediction module $\mathcal{F}$ is the real Fourier transform (FT) of the state sequences generated by the optimal policy $\pi^*$. **According to the last inequality of the proof provided in the global response, the distance between the predicted FT and the ground truth is constrained by the objective of our method (SPF).** This can be expressed as follows:
> > >   $$
> > >   \begin{align}
> > >   \| F_{\pi_*} - F_{\overline{\pi}\circ\phi}\|
> > >       \leq (1-\gamma)^{-1}\cdot E_{(s,a,s')\sim\mathcal{D}}\left[s'+e^{-j\omega}\gamma E_{a'\sim{\overline{\pi}\circ\phi}}\left[F_{\overline{\pi}\circ\phi}(s',a')\right] - F_{\overline{\pi}\circ\phi}(s,a)\right].
> > >     \end{align}
> > >   $$
> > >   Here, $F_{\pi_*}$ represents the real FT of the state sequence generated by the optimal policy $\pi_*$, and $F_{\overline{\pi}\circ\phi}$ represents the output of the prediction module $\mathcal{F}$, which is influenced by the current policy $\overline{\pi}$ and the current encoder $\phi $.
> > >
> > >   According to the inequality we mentioned above, the distance between the predicted FT of our module $\mathcal{F}$ and the real FT is minimized when we minimize our objective $\overline{\delta}\_\pi(s,a,s'):=s'+e^{-j\omega}\gamma E\_{a'\sim{\pi}}\left[F\_{\pi}(s',a')\right] - F\_{\pi}(s,a)$ and collect a sufficient number of samples in the replay buffer $\mathcal{D}$.
> > >
> > > + **We present a comparison table between the real Fourier transform (FT) and the predicted FT.** To obtain these results, we use our trained policies to interact with six MuJoCo environments and record the states and the actions for the 201 last steps of an episode, denoted as $\\{s_t\\}\_{t=0}^{200}$ and $\\{a_t\\}\_{t=0}^{200}$, respectively. **The real FT** of the state sequence $\\{s_1,s_2,\dots,s_{200}\\}$ over a 200-step horizon is computed using the formulation of the Discrete-time Fourier transform (DTFT).  For the predicted FT, we employ our trained encoder and prediction module, with the state $s_0$ and the action $a_0$ serving as inputs. The output of the prediction module represents **the predicted FT** of the state sequence $\\{s_1,s_2,\dots,s_{200}\\}$.  We compute the cosine similarity between $A$ and $B$ with the formula $-\frac{A\cdot B}{\\|A\\|\\,\\|B\\|}$, where values closer to $-1$ indicate greater similarity while values closer to $1$ indicate greater dissimilarity. The cosine similarity between the real FT and the predicted FT is listed below.
> > >
> > > | Environments | Cosine similarity between the real FT and the predicted FT |
> > > | ------------ | ---------------------------------------------------------- |
> > > | HalfCheetah  | -0.655                                                     |
> > > | Hopper       | -0.660                                                     |
> > > | Walker2d     | -0.651                                                     |
> > > | Ant          | -0.351                                                     |
> > > | Swimmer      | -0.557                                                     |
> > > | Humanoid     | -0.651                                                     |
> > >
> > >
> > >
> > >
> > > **Q2. The existence of $\frac{R_\text{max}}{1-\gamma}$ in Theorem 1.**
> > >
> > > **A2.**
> > >
> > >
> > > Thank you for raising this point! **We acknowledge that when $\frac{R_\text{max}}{1-\gamma}$ takes on a large value, our claim that state sequences exhibit significant differences in the presence of a significant difference in policy performance is weakened.** An alternative viewpoint regarding Theorem 1 is that as the distribution of state sequences generated by the trained latent policy $\overline{\pi}\circ\phi$ approaches the distribution of state sequences generated by the optimal policy $\pi^*$, the performance of our trained policy also approaches that of the optimal policy.

---

### Official Review · Reviewer_KaZ7 · 2023-07-04

**Soundness:** 1 poor
**Presentation:** 2 fair
**Contribution:** 2 fair
**Rating:** 5
**Confidence:** 3

**Summary:**

Data-efficient RL exploits representation learning which learns representations by predicting long-term future states. However, inherent structural information in sequential state signals is ignored in existing methods. The authors propose SPF which exploits the frequency domain of state sequences. The proposed method performs an auxiliary self-supervision task to improve the efficiency of representation learning. This method outperforms several state-of-the-art algorithms in terms of both sample efficiency and performance.

**Strengths:**

* Notations are clear and methods are clearly proposed.


**Weaknesses:**

* Results are not promising compared to SAC-OFE. In Humanoid, SAC-OFE outperforms SAC-SPF, and in HalfCheetah, Walker2d, Ant, Swimmer, SAC-OFE is comparable to SAC-SPF.
* I could not find critical superior points to use Fourier transform. As the authors pointed out in the conclusion, experiments on visual-based RL settings are required to show the benefits of the Fourier transform.
* The inherent structural information in a sequential state is not clearly discussed in the paper.


**Minors**
* Typo in Figure2, There is no predictor F in the figure.



**Questions:**

* What is inherent structural information we can get from Fourier-transform?

* The main concern is that results do not outperform the baseline. Also, in 6.3, the results of all the baselines and SPF are comparable.

====================================================================================================


The authors successfully handle my concerns. I raise my score to borderline accept.

**Limitations:**

Yes

---

> ### Author Rebuttal · Authors · 2023-08-10
>
> We thank the reviewer for the insightful and valuable comments. We respond to your comments as follows and sincerely hope that our rebuttal could properly address your concerns. If so, we would deeply appreciate it if you could raise your score（"3: Reject"). If not, please let us know your further concerns, and we will continue actively responding to your comments and improving our submission.
>
> **Figures S1, S2, and Tables S1, S2 in this response refer to the pdf attached in the global response.**
>
>
> **Q1. Results are not promising compared to SAC-OFE.**
>
> **A1.**
> + **Our method (SPF) learns more quickly than OFE and significantly outperforms OFE by a large margin when the number of interactions is limited.** It is particularly important when we train the agent under a constrained budget for interactions, which can be costly. In Table S1 and Figure S1(a) of **the attached pdf**, we report that for **MuJoCo tasks**, SPF achieves **a 39% and 18% boost on average** compared to OFE at 500k and 1M interaction steps, respectively.
>
> + **We conduct experiments to test our method (SPF) on complex dexterous manipulation tasks, the Adroit benchmark [1].** The Adroit tasks are difficult to solve effectively due to their high dimensionality, which provides an opportunity to highlight the advantage of SPF in representation learning. In Table S1 and Figure S1(b) of the pdf, we report that for **Adroit tasks**, SPF achieves **a 1494% and 330% boost on average** compared to OFE at 1M and 2M interaction steps, respectively.
>
>   For your convenience, we quote the **Adroit-1M** section of Table S1 below.
>
> | Adroit-1M      | SAC-SPF  | SAC-OFE | PPO-SPF  | PPO-OFE |
> | - | - | - | - | - |
> | AdroitHandDoor | **4420** | 1162    | -56      | -58     |
> | AdroitHandPen  | **960**  | -547    | **1453** | -253    |
>
> **Q2. Results in Figure 6.3 are comparable.**
>
> **A2. Our method (SPF) learns more quickly than the other baselines and significantly outperforms other baselines when the number of interactions is limited.** Figure S2(a) in the added pdf displays the learning curves on HalfCheetah and Walker2d tasks for the initial 1M interaction steps, providing evidence of SPF's improved performance in terms of sample efficiency.
>
> **Q3. Experiments on visual-based RL to show the benefits of the FT**.
>
> **A3. We extend the evaluation of SPF to the visual-based RL setting** using DeepMind Control Suite (DMC), a popular benchmark for visual-based RL.  We build our representation learning method (SPF) on DrQ [2], an existing visual RL approach that improves SAC with image augmentation. We then compare the performance of DrQ+SPF against the original DrQ on DMC. We use a mini-batch size of $128$ and an action repeat of $4$ to accelerate the training speed. In Table S2 of the pdf, we report that for the **Finger-spin and Walker-walk tasks**, DrQ+SPF achieves **a 2% and 25% boost on average** compared to the original DrQ at 100k and 500k interaction steps, respectively.
>
> For your convenience, we quote the **DMC-500k** section of Table S2 below.
>
> | DMC-500k | DrQ+SPF | DrQ  |
> | - | - | - |
> | Finger spin | **728** | 543 |
> | Walker walk | **860** | 743 |
>
> **Q4. The critical superior points of using FT.**
>
> **A4.**
> + Intuitively, we train the encoder by predicting the Fourier Transform (FT) of state sequences to **extract the periodic patterns exhibited by state sequences in the long run, which are valuable for long-term decision-making RL tasks**. In Theorem 2, we show the widespread existence of asymptotically periodic behaviors in a Markov chain, where we can fully exploit the potential of FT to extract the regularity information.
>
> + The formulation of Discrete-Time FT (DTFT) offers **an elegant prediction loss in the form of TD error** (See Equation (6) and equation on Line 261). The TD-style loss makes SPF **easy to implement and eliminates the need to store multi-step future states as labels for predicting future state sequences.**
> + **The FT of state sequences preserves the state sequence's ability to reflect the performance of the current policy.** Theorems 1 and 3 provide theoretical evidence for the relationship between policy performance and state sequences in the time and frequency domains, respectively.
>
> **Q5. What is inherent structural information we can get from FT?**
>
> **A5.** In our paper, the two types of structural information mean **the policy performance information reflected by state sequences** and **the regularity information inherent in state sequences**. We provide detailed explanations as follows.
>
> + The first information refers to the dependency between reward sequences and state sequences shown in Figure 1 of the main text, where the state sequences implicitly reflect the performance of the current policy (i.e. the discounted sum of a reward sequence) and exhibit significant differences under good and bad policies (Theorem 1). Theorem 3 showd that the FT of state sequences preserves a similar property related to policy performance, which means we can get information of policy performance from the FT of state sequences.
> + The second information refers to the temporal dependencies in a state sequence, namely the regularity patterns exhibited by the state sequence (showed in Theorem 2 and visualized in Figures 5-9 of the Appendix). By analyzing the FT, we can identify the dominant frequencies of the time-horizon state sequences, which reveal the periodic patterns within the state sequences. In Figure 4(c) of the main text, we demonstrate that the representations learned by SPF effectively capture such regularity information in state sequences.
>
> **Q6. Typo in Figure 2.**
>
> **A6.** Thank you for pointing out this typo. In Figure 2 of the main text, the letter $\mathcal{P}$ should be $\mathcal{F}$.
>
>
>
> [1] Learning complex dexterous manipulation with deep reinforcement learning and demonstrations.  RSS, 2018.
>
> [2]  Image augmentation is all you need: Regularizing deep reinforcement learning from pixels. ICLR, 2021.

---

### Official Review · Reviewer_ov12 · 2023-07-08

**Soundness:** 4 excellent
**Presentation:** 4 excellent
**Contribution:** 4 excellent
**Rating:** 8
**Confidence:** 3

**Summary:**

Unlike many prior model-based RL methods that predict one future state at a time, the authors present a Fourier-transform based approach that predicts future states over a fixed interval. They prove that a large difference in policy performance will cause a significant difference between the experienced states of a good and bad policy. This conclusion is important for utilizing frequency-space analysis of the states.

Their approach uses a self-supervised learning objective where they attempt to fit an online encoder given the current states, and a target encoder given the next states. Note that both the online and target encoder, given a single state, will predict frequencies for the rest of the trajectory. This is in contrast to prior methods that will just compute the next state. The TD-like error between predictions is then used to train the online encoder. I'm not sure I fully understand it, but it seems quite elegant in that their approach closely resembles the one-step TD error. As far as I can tell, they are not actually using the predicted state sequences for planning, as in model-based RL. Rather, they are learning the features necessary to predict future state sequences. These features are useful for a downstream policy.

They evaluate their method on the standard MuJoCo/Gym benchmarks which outperforms baselines. They compare their method using SAC/PPO to SAC/PPO using OFE or no prediction at all. They continue to ablate nearly all portions of their approach.

**Strengths:**

- The paper is very well written
- Their approach is novel and elegant
- Their experiments, specifically the ablation, is well-done
- They show their approach outperforms baselines

**Weaknesses:**

- They only compared to one other feature-learning algorithm. It would be beneficial to have additional baselines.
- Unless I'm misunderstanding, it seems like they've purposely limited the applications of their approach to fully observable, model-free RL
    - They already predict future states, they might as do some online planning
    - This could be applied to partially observable domains to learn Markov states
- They do not list any limitations of their approach

**Questions:**

- "Therefore, state sequences maximally preserve the influence of the transition intrinsic to the environment and the effect of actions generated from the current policy"
    - I'd be careful saying this. In the case of a deterministic policy, yes. But if you add stochasticity (e.g. epsilon greedy or policy-gradient sampling) I'm not sure you can claim you can predict a future state sequence. You can perhaps predict the distribution of states.
- "it is widely accepted that the frequency domain shows the regularity properties of the time-series data"
    - There are citations for this but I find it a bit hard to believe. For cyclical processes this makes sense, but there is no guarantee, for example, that you will see a specific feature every k timesteps. Could you add an intuitive explanation of why this should work for latent state sequences?
    - This is explained later in the paper, but perhaps it makes sense to say you show/prove this in the introduction
- Eq. 5: Why do you discount the future state using gamma? I understand dicounting the reward/return, but not effectively "decaying" the future state. Could you briefly explain the reasoning here? Is it to weight nearby states more heavily?

**Limitations:**

- The authors do not list any limitations

---

> ### Author Rebuttal · Authors · 2023-08-10
>
> We thank the reviewer for the positive and insightful comments. We respond to each comment as follows and sincerely hope that our rebuttal could properly address your concerns. If so, we would deeply appreciate it if you could raise your score. If not, please let us know your further concerns, and we will continue actively responding to your comments and improving our submission.
>
> **Figure S2 and Table S3 in this response refer to the pdf attached in the global response.**
>
> **Q1. Additional baselines.**
>
> **A1. We further compare our method (SPF) with an additional baseline D2RL [1]**, which explores an effective architectural of deeper networks and dense connections for representation learning. The evaluation data of D2RL on state-based RL tasks is sourced from the 'Deeper_Larger_Actor-Critic_RL' repository on GitHub. In Table S3 of the added pdf, we report that for **five MuJoCo tasks**, SPF achieves **a 39% and 18% boost on average** compared to the other baselines at 500k and 1M interaction steps, respectively.
>
> For your convenience, we quote the **MuJoCo-1M** section of Table S3 below.
>
> | MuJoCo-1M   | SAC-SPF   | SAC-OFE  | SAC-D2RL | PPO-SPF  | PPO-OFE | PPO-D2RL |
> | - | - | - | - | - | - | - |
> | HalfCheetah | **15822** | 14425    | 12932    | **3154** | 3066    | 1592     |
> | Hopper      | **3517**  | 3197     | 3247     | **3152** | 2370    | 1698     |
> | Walker2d    | **5158**  | 4833     | 5075     | **2229** | 1080    | 1596     |
> | Ant         | **7241**  | 6738     | 5385     | **1053** | 913     | 236      |
> | Humanoid    | 5633      | **6241** | 5073     | 441      | **448** | 413      |
>
> **Q2. Apply their approach to partially observable domains to learn Markov states.**
>
> **A2. We extend the evaluation of SPF to partially observable domains** by introducing standard Gaussian noises to substitute several dimensions of the input states. Specifically, we mask 2 and 3 dimensions of states and evaluate the performance of our method in comparison to other baselines on **Walker2d task**, which is one of the MuJoCo tasks. Figure S2(b) of the added pdf in 'global' response demonstrates the superiority of our method over OFE and raw RL algorithms.
>
>
>
> **Q3. They might as do some online planning.**
>
> **A3.** It is possible to apply SPF to online planning by planning through the predicted state sequences. For instance, in the AdroitHandDoor task, where we aim to teach a robotic arm to open a door. The Fourier functions of the state sequences predicted by SPF can be recovered to obtain the original state sequences using the inverse Fourier Transform. This enables us to extract the physical parameters of the robotic arm and the coordinates of the door handle from the states. Subsequently, optimization or heuristic methods can be employed to generate a path for the robotic arm to reach the door handle.
>
>
>
> **Q4. Lack of limitations.**
>
> **A4.** Thank you for pointing out this.
>
> + One of the main limitations of our paper is the absence of an analysis regarding the performance of our method on tasks that lack periodicity. In the revised version, we will thoroughly analyze the applicability of our approach to non-periodic tasks. Additionally, we will investigate its ability to identify irregular changes in time-series data.
> + Another limitation of our study is that we did not extend our proposed method to visual-based RL settings, where frequency domain transformations of images have shown promise for noise filtering and feature extraction.
>
>
>
> **Q5. If you add stochasticity, I'm not sure you can claim you can predict a future state sequence.**
>
> **A5.** In theoretical analysis, we model the state sequences as the expectation of state sequences with respect to a stochastic policy (refer to Equation (5)). Hence, in theory, we account for the inherent randomness associated with the policy. In practical implementation, we optimize the objective function$\overline{\delta}_\pi(s,a,s'):=s'+e^{-j\omega}\gamma E\_{a'\sim{\pi}}\left[F\_{\pi}(s',a')\right] - F\_{\pi}(s,a)$. Note that the selection of the next action $a'$ is based on the current policy $\pi$, which implies that the prediction of the Fourier Transform incorporates the stochasticity of the policy.
>
>
>
> **Q6. Why this should work for latent state sequences ?**
>
> **A6.** Thank you for bringing this to our attention. **The learned Fourier feature may contain frequency information about the future state sequences, but it does not imply that the feature itself possesses periodic properties.** For example, we assume that each state corresponds to one latent state, and it is possible for multiple states to correspond to the same latent state, which we refer to as an equivalence class consisting of these states.  Consequently, the periodicity of the latent states may be determined by the least common multiple of the states within the corresponding equivalence class. This situation potentially disrupts the inherent periodicity of the latent states.
>
>
>
> **Q7. About Equation (5).**
>
>
> **A7.** Since we consider the infinite state sequences, it is necessary to keep the Fourier series remains bounded or converges to a finite value. And the discounted rate for state sequences can ensure the recursive operator $\mathcal{T}$ in Equation (7) a contraction mapping, which guarantees the convergence to the true FT function in tabular settings when using our auxiliary loss. This modeling approach has been used in prior works [2], which allows us to compactly express the difference in performance between two policies.
>
>
>
> [1] D2rl: Deep dense architectures in reinforcement learning. arXiv preprint, 2020.
>
> [2] Achiam J, et al. Constrained policy optimization, ICML 2017.

---

> > ### Comment · Reviewer_ov12 · 2023-08-14
> >
> > Thank you for adding the additional baselines and fully addressing my concerns.
> >
> > I was unclear in my comment on partial observability and online planning, I apologize. My understanding is that your approach could allow predicting future states without sequentially rolling them out. For example, if I wanted to predict $s_{t + 100}$ using traditional MBRL, I would need to loop through my dynamics model 100 times, potentially accruing error along the way. Your current approach conditions the model on all future actions (Eq. 5), but if it instead conditioned on the current action and policy (i.e., $s_{t+L} = g(s_t, a_t, L | \pi)$, one could effectively predict $s_{t + 100}$ in one forward pass, right? Or perhaps I am misunderstanding.
> >
> > This seems like it could be a powerful property if I were training a Dreamer-style model with partial observability, as one could learn a latent state representation $s_t = f(o_1, \dots, o_t)$ that predicts future observations via $g(s_t, a_t, 100,000 | \pi) = o_{t+100,000} $, without worrying about computational cost. Additionally, one could do fast online planning of long trajectories that would not be possible with RNN or transformer-based dynamics models.

---

> > > ### Author Response · Authors · 2023-08-16
> > > **Thanks for your comments**
> > >
> > > Thank you for the insightful comment! **We believe that our method (SPF) can do fast online planning in a Dreamer-style algorithm, in which SPF could generate multi-step future states without using RNN or transformer-based dynamics models.** The reasons are as follows.
> > >
> > > + Our method could predict future states without sequentially rolling them out. However, **we would like to point out that our method conditions the prediction model on the current state $s_t$ and the current action $a_t$, rather than all future actions.** According to Equation (5) in our main text, we predict the expectation of the future state sequences $\\{s_{t+1}, s_{t+2}, \cdots\\}$ conditioned on the current state $s_t$ and the current action $a_t$. Note that the expectations in Equations (5) and (7) are computed with respect to the policy $\pi$ and the dynamic $p$, indicating that the predicted state sequences are also conditioned on the policy.
> > > + In practical implementation, the inputs of the predictor module are the representations of the current state $s_t$ and the current action $a_t$. The output of the predictor is an $L-$​dimensional vector, which corresponds to $L$ discrete samples of the discrete-time Fourier transform (DTFT) of the future state sequences. According to the inverse discrete Fourier transform (IDFT), we can compute the future state sequences from the predicted DTFT, thereby enabling the estimation of $s\_{t+L}$ for $L=1,2,\cdots,H$ in one forward pass. **In other words, by leveraging the IDFT, the Fourier predictor can generate multi-step future states for fast online planning.**

---

### Official Review · Reviewer_1q5C · 2023-07-13

**Soundness:** 3 good
**Presentation:** 3 good
**Contribution:** 3 good
**Rating:** 6
**Confidence:** 4

**Summary:**

The paper suggests a representation learning method that leverages the existing periodic structures in the state sequences of a policy. Specifically, they propose using the features obtained by the Fourier transform of the state sequence. To practically extract such features, they employ a Bellman equation-like recursive form of the Fourier transform (FT), which enables efficient online learning. The authors demonstrate that their representation learning method helps improve the performances of SAC and PPO in Gym MuJoCo environments.

**Strengths:**

- The idea of using frequency features in sequential decision-making problems seems novel and sensible, given that real-world tasks often inherently have periodic (sub-)structures in their states (e.g., robotic locomotion).
- The recursive objective in Equation (7), which enables learning Fourier features in practice without the need to deal with the entire state sequences, is intriguing and potentially inspirational for future research.
- The paper includes an informative ablation study.

**Weaknesses:**

- The Fourier features inherently depend on the policy and task, making them less reusable for multiple tasks, especially compared to previous self-supervised representation learning methods.
- The theorems in the paper do not necessarily explain why Fourier features could be potentially useful for *representations*. Theorem 3 states a (very crude) bound for the performance difference between two policies expressed in the frequency domain. However, it is unclear as to why the features extracted in this manner are useful or sufficient for modeling policies/value functions as well, which is the way SPF utilizes the learned features. I would have expected a theorem that says the sub-optimality of using the learned Fourier representation decreases as the state sequence shows stronger periodicity.


**Questions:**

- Can the authors elaborate on why Fourier features are useful for representations? For example, let's say a state sequence from a policy forms a perfect sinusoidal curve so that its Fourier transform collapses to a single point. In such a case, its Fourier representation (which is a constant) may not contain sufficient information for control (i.e., for producing optimal actions to maintain that sinusoidal curve).
- Could SPF be harmful in tasks that do not have periodicity? Have the authors tested SPF on such environments/tasks?


**Limitations:**

The authors briefly mention the limitations of SPF in Section 7. However, it would be also important to include a discussion on the second question above in the limitation section as well.

---

> ### Author Rebuttal · Authors · 2023-08-10
>
> We thank the reviewer for the positive and insightful comments. We respond to each comment as follows and sincerely hope that our rebuttal could properly address your concerns. If so, we would deeply appreciate it if you could raise your score. If not, please let us know your further concerns, and we will continue actively responding to your comments and improving our submission.
>
> **Figure S1 and Table S1 in this response refer to the pdf attached in the global response.**
>
> **Q1. Could SPF be harmful in tasks that do not have periodicity? Have the authors tested SPF on such tasks?**
>
> **A1.**
> + **We claim that SPF shows favorable performance even in tasks lacking periodicity.** By analyzing the Fourier Transform (FT), we can identify the dominant frequencies and their corresponding magnitudes, revealing the periodic patterns or oscillatory behavior within the time-horizon state sequences. Furthermore, the Fourier Transform enables the identification of transient phenomena, such as irregular sudden changes, which may be reflected in the frequency domain as high-frequency components.
>
> + We further test SPF on complex dexterous manipulation tasks, **the Adroit benchmarks** [1], which **do not have clear periodicity due to the randomized position of the manipulated target** (such as the door in the AdoritHandDoor task). In Table S1 and Figure S1(b) of **the attached pdf in 'global' response**, we report that for **Adroit tasks**, SPF achieves **a 1494% and 330% boost on average** compared to OFE at 1M and 2M interaction steps, respectively.
>
>   For your convenience, we quote the **Adroit-1M** section of Table S1 below.
>
> | Adroit-1M      | SAC-SPF  | SAC-OFE | PPO-SPF  | PPO-OFE |
> | - | - | - | - | - |
> | AdroitHandDoor | **4420** | 1162    | -56      | -58     |
> | AdroitHandPen  | **960**  | -547    | **1453** | -253    |
>
> **Q2. A theorem that says the sub-optimality of using the learned Fourier representation decreases as the state sequence shows stronger periodicity is expected.**
>
> **A2.**
> + We provide **a sub-optimality performance bound for the latent policy using the learned Fourier representation** as follows:
>
>   *Suppose that $\mathcal{S}\subset\mathbb{R}^D$ the reward function $R(s,a,s')=R(s)$ is a linear function with respect to $s\in \mathcal{S}$. Then the performance difference between the optimal policy* $\pi^*$ *and the latent policy* $\overline{\pi}\circ \phi $ *may be bounded as:*
>   $$
>   |J(\pi^*)-J(\overline{\pi}\circ\Phi)|\leq \frac{\sqrt{D}\cdot R_\text{max}}{(1-\gamma)^2}\cdot E_{(s,a,s')\sim\mathcal{D}}\left[\overline{\delta}_{\overline{\pi}\circ\phi }(s,a,s')\right],
>   $$
>
>   *where $\overline{\delta}_\pi(s,a,s'):=s'+e^{-j\omega}\gamma E\_{a'\sim{\pi}}\left[F\_{\pi}(s',a')\right] - F\_{\pi}(s,a)$ is the minimized objective of SPF.*
>
>   **We provide detailed proof in the 'global' response.**
>
> + Based on the above theorem, the performance gap between the optimal policy and our learned policy decreases when we minimize the auxiliary loss function $\overline{\delta}_\pi(s,a,s')$, even in cases where the state sequence does not exhibit strong periodicity. The above theorem indicates that **leveraging Fourier spectrum information via the objective of SPF is useful for the agent to approach the optimal policy.**
>
> + However, **it has benefits when the state sequence shows stronger periodicity.** In such scenarios, the FT of the state sequence approaches discreteness, which reduces errors introduced during algorithm implementation. This is because, in practice, we can only estimate the values of Fourier functions from discrete sampling points.
>
>
>
> **Q3. Why are Fourier features useful for representations?**
>
> **A3.**
> + The inverse Fourier Transform allows us to convert the frequency domain back into the time domain. When given a single frequency $\omega$, we can express the sinusoidal function as $\sin(\omega x)$ (if we do not consider the amplitude and phase factors).
>
> + In the case of a perfect sinusoidal curve, as the representation learns a Fourier function on the interval of $[0,2\pi]$,  it may discover that there exists a single frequency $\omega$ on $[0,2\pi]$, and **all values outside the point (that is, $[0,2\pi]\setminus \\{\omega\\}$) are zero**. Then the representation captures the presence of the perfect sinusoidal function.
>
>
>
> **Q4. More discussion for limitations.**
>
> **A4.** Thank you for pointing out this. One of the main limitations of our paper is the absence of an analysis regarding the performance of our method on tasks that lack periodicity. In the revised version, we will thoroughly analyze the applicability of our approach to non-periodic tasks. Additionally, we will investigate its ability to identify irregular changes in time-series data.
>
>
>
> [1]. Rajeswaran A, et al. "Learning complex dexterous manipulation with deep reinforcement learning and demonstrations." RSS, 2018.

---

> > ### Comment · Reviewer_1q5C · 2023-08-10
> > **Response to Authors**
> >
> > While I greatly appreciate the new results on Adroit, they have left me rather confused about the effectiveness of Fourier features. If there is no periodicity in trajectories, why would Fourier features be helpful? Perhaps an alternative hypothesis is that the performance gain comes *not* from Fourier features but from some practical implementation details, hyperparameters, etc., that simply lead to better performance or make the representations better. I also acknowledge that this benefit might come from detecting irregular, sudden changes (as the authors mentioned), but it would require more controlled analyses to prove this point (e.g., Do the learned Fourier features actually detect such sudden changes?, Why detecting such events is necessarily helpful for performance?, etc.). That being said, I wouldn't decrease my score as I believe the ideas and initial results are still valuable.
> >
> > Regarding the perfect sinusoidal curve example, my point is that since the Fourier representation $z$ is always a *constant* (i.e., always only non-zero at $\omega$), it might be insufficient for the *policy* $\pi(a|z)$. How can a policy produce optimal actions when the variable on which it is conditioned is always a constant?

---

> > > ### Author Response · Authors · 2023-08-13
> > > **Thanks for your comments**
> > >
> > > Thank you for your insightful comments and for helping us gain deeper insights into our work. We respond to your comments as follows and sincerely hope that our response could properly address your concerns.
> > >
> > > **Q1. About the effectiveness of Fourier features on Adroit tasks.**
> > >
> > > **A1.**
> > >
> > > Thank you for raising this point. By outputting the state sequences generated by our learned policy, we observed that the AdroitHandDoor task exhibits stronger periodicity compared to the AdroitHandPen task. This observation may help explain why our method performs more effectively in the Door task compared to the Pen task. **According to the properties of the Fourier transform, our method demonstrates better performance on tasks with strong periodicity and has the ability to capture sudden changes with periodicity.**
> > >
> > >
> > >
> > > **Q2: About the perfect sinusoidal curve example.**
> > >
> > > **A2.**
> > >
> > > We apologize for the misunderstanding. Here is our explanation. In the practical implementation of our method, we predict the Fourier transform (FT) of the future state sequence $\\{ s_{t+1},s_{t+2}, \cdots \\}$ based on the current state $ s_t$ and the current action $a_t$. Specifically, **for different states along the sinusoidal curves, the corresponding future state sequences start from different time steps, resulting in sinusoidal curves with varying time shifts, i.e., different phases.** Thus, in the case of perfect sinusoidal curves, the future state sequence predicted from state $s_t$ and action $a_t$ can be expressed as follows:
> > > $$
> > > f(t)=\begin{cases}
> > > sin(\omega_0 t+\varphi)& \text{t}>0, \\
> > > 0& \text{t}\leq 0
> > > \end{cases}
> > > =sin(\omega_0 t+\varphi) H(t),
> > > $$
> > > where $\omega_0$ means the frequency, $\varphi$ means the phase, and $H(t)$ means the Heaviside step function. The Heaviside step function takes on a value of zero for negative arguments and a value of one for positive arguments.
> > >
> > > According to the convolution theorem, the Fourier transform of $f(t)$ is given by:
> > > $$
> > > F(\omega)=-\cos \varphi \cdot\frac{\omega_0}{\omega^2-\omega_0^2}+\frac{\pi}{2}\sin \varphi \left( \delta(\omega-\omega_0)+\delta(\omega+\omega_0) \right)-i\left[ \sin \varphi \cdot\frac{\omega}{\omega^2-\omega_0^2}+\frac{\pi}{2}\cos \varphi\left( \delta(\omega-\omega_0)+\delta(\omega+\omega_0) \right) \right],
> > > $$
> > > where we use Fourier transform in the non-unitary form with angular frequency.
> > >
> > > Note that the phase $\varphi$ plays a role in determining the FT values of future state sequences that start from different time steps. The encoder and the predictor capture such phase information when predicting the FT of the future state sequences. Thus, **the representations of the states along the sinusoidal curves can be distinguished by their respective phases.** We agree that the states spaced one cycle apart share the same representations, which means the agent exhibits periodic patterns of behavior and performs the same actions selected by the policy $\pi(\cdot|\phi(s))$ when encountering these states with the same phase.
> > >
> > > **We recognize that the influence of the phase $\varphi$ may be relatively weaker compared to the dominant frequency, as influenced by the delta function.**  Therefore, in practical implementations, we address this by either concatenating the original state with Fourier features or training the encoder using the critic loss and prediction loss simultaneously.

---

> > > > ### Comment · Reviewer_1q5C · 2023-08-14
> > > >
> > > > Thank you for the detailed clarification on the perfect sinusoidal curve example!

---

> > > > > ### Author Response · Authors · 2023-08-16
> > > > > **Thank you for your kind support**
> > > > >
> > > > > Thank you again for your valuable comments and constructive suggestions, which are of great help to improve the quality of our work. We sincerely hope that our rebuttal could properly address your concerns. If you have further concerns, please let us know, and we will continue actively responding to your comments and improving our submission.

---

### Author Rebuttal · Authors · 2023-08-10

We thank all the reviewers for the insightful comments and constructive suggestions, which are very helpful for us to strengthen this submission.

We have submitted detailed responses to address the concerns raised by all the reviewers. Besides answering the technical questions,

1. We provide a sub-optimality performance bound for the latent policy using the learned Fourier representation, which narrows the gap between the motivation and the implementation and explains why the learned Fourier features could be potentially useful for representations; (for reviewers 1q5C and  i4e2)
2. We conduct experiments to test our method on complex dexterous manipulation tasks, the Adroit benchmark, which demonstrates the effectiveness of our representation method to deal with high-dimensional inputs and non-periodic tasks; (for reviewers 1q5C and KaZ7)
3.  We extend the evaluation of our method to the visual-based RL setting on DeepMind Control tasks, which demonstrates the benefits of the Fourier transform on image feature extraction; (for reviewer KaZ7)
4. We provide more detailed results with the performance of our method at limited interaction steps, which demonstrates the superiority of our method in terms of sample efficiency; (for reviewer KaZ7)
5. We compare our method with an additional baseline D2RL, which demonstrates the superiority of our method over other feature-learning algorithms; (for reviewer ov12)
6. We extend the evaluation of SPF to partially observable domains, which demonstrates the effectiveness of our method with a lack of information. (for reviewer ov12)

We would like to know if our responses have properly addressed your concerns. All of your feedback and/or additional comments are warmly welcomed.

**We provide the proof of the bounded sub-optimality theorem as follows.**

Based on the proof of Lemma 1 in [1], we can prove a similar lemma as follows.

*Let Fourier transform function $F_\pi:=\mathcal{F}\widetilde{s}_{0}(\omega|\pi)$. For any policy $\pi$ and any function $f:S\rightarrow \mathbb{R}$, the following formula holds:*

$$
E_{s\sim\mu}[f(s)] +  \frac{1}{1-\gamma}E_{\substack{s\sim c^\pi \\ a\sim\pi \\ s'\sim P}}\left[\delta(s,a,s')\right] = F_\pi,
$$

*where*  $c^\pi(s)=(1-\gamma)\sum_{t=0}^\infty e^{-j\omega t}\gamma^tP(s_t=s|\pi,\mathcal{M})$,  $F\_\pi(s):=\mathcal{F}\widetilde{s}\_{0}(\omega|\mathbf{s_0}=s,\pi)\in\mathbb{C}^D$ *and* $\delta(s,a,s'):=s'+e^{-j\omega}\gamma f(s') - f(s)$.

Apply any two policy $\pi_1$ and $\pi_2$ to the above and let $f(s)=F_{\pi_2}(s)$ in the two obtained equations, then we get

$$
E_{s\sim\mu}[F_{\pi_2}(s)] +  \frac{1}{1-\gamma}E_{\substack{s\sim c^{\pi_1} \\ a\sim{\pi_1} \\ s'\sim P}}\left[s'+e^{-j\omega}\gamma F_{\pi_2}(s') - F_{\pi_2}(s)\right] = F_{\pi_1},
$$

and

$$
E_{s\sim\mu}[F_{\pi_2}(s)] +  \frac{1}{1-\gamma}E_{\substack{s\sim c^{\pi_2} \\ a\sim{\pi_2} \\ s'\sim P}}\left[s'+e^{-j\omega}\gamma F_{\pi_2}(s') - F_{\pi_2}(s)\right] = F_{\pi_2}.
$$

Take the difference between the above two equations, then we have

$$
F_{\pi_1} - F_{\pi_2} = (1-\gamma)^{-1}\cdot \left( E_{\substack{s\sim c^{\pi_1} \\ a\sim{\pi_1} \\ s'\sim P}}\left[\delta_{\pi_2}(s,a,s')\right] - E_{\substack{s\sim c^{\pi_2} \\ a\sim{\pi_2} \\ s'\sim P}}\left[ \delta_{\pi_2}(s,a,s')\right]\right).
$$
Note that the second term within the parentheses is zero, which is easy to check from the summation formulation of the Fourier transform $F_\pi$. Thus, we have
$$
F_{\pi_1} - F_{\pi_2} = (1-\gamma)^{-1}\cdot E_{\substack{s\sim c^{\pi_1} \\ a\sim{\pi_1} \\ s'\sim P}}\left[\delta_{\pi_2}(s,a,s')\right].
$$

Replacing $\pi_1$ and $\pi_2$ in the above equation with the optimal policy $\pi_*$ and the latent policy with the learned representation $\overline{\pi}\circ\phi$, respectively. Then their performance difference can be bounded as follows.

$$
\begin{align}
\| F_{\pi_*} - F_{\overline{\pi}\circ\phi}\|
    &\leq (1-\gamma)^{-1}\cdot E_{\substack{s\sim c^{\pi_*} \\ a\sim{\pi_*} \\ s'\sim P}}\left[\delta_{\overline{\pi}\circ\phi}(s,a,s')\right] \\
    &= (1-\gamma)^{-1}\cdot E_{\substack{s\sim c^{\pi_*} \\ a\sim{\pi_*} \\ s'\sim P}}\left[s'+e^{-j\omega}\gamma E_{a'\sim{\overline{\pi}\circ\phi}}\left[F_{\overline{\pi}\circ\phi}(s',a')\right] - E_{a\sim{\overline{\pi}\circ\phi}}\left[F_{\overline{\pi}\circ\phi}(s,a)\right]\right] \\
    &\leq (1-\gamma)^{-1}\cdot \max\limits_{s,a} E_{s'\sim p(\cdot|s,a)}\left[s'+e^{-j\omega}\gamma E_{a'\sim{\overline{\pi}\circ\phi}}\left[F_{\overline{\pi}\circ\phi}(s',a')\right] - F_{\overline{\pi}\circ\phi}(s,a)\right] \\
    &\approx (1-\gamma)^{-1}\cdot E_{(s,a,s')\sim\mathcal{D}}\left[s'+e^{-j\omega}\gamma E_{a'\sim{\overline{\pi}\circ\phi}}\left[F_{\overline{\pi}\circ\phi}(s',a')\right] - F_{\overline{\pi}\circ\phi}(s,a)\right].
   \end{align}
$$

Let $\overline{\delta}\_\pi(s,a,s'):=s'+e^{-j\omega}\gamma E\_{a'\sim{\pi}}\left[F\_{\pi}(s',a')\right] - F\_{\pi}(s,a)$ and substitute the above inequation into Theorem 3, then we achieve the desired bound.

[1] Achiam J, et al. Constrained policy optimization, ICML 2017.

---

### Decision · Program_Chairs · 2023-09-21

**Decision:**

Accept (spotlight)

**Comment:**

One of the shortcoming of prior predictive coding auxiliary task works in RL is prediction in the time domain, where relevant structure may be difficult to extract (e.g., using sequential state representation prediction). This work presents a novel and interesting approach formulating an auxiliary task in the frequency domain. This bridges modern auxiliary tasks in RL with prior works that attempted to formulate value learning in the frequency domain, an interesting and relevant outcome. The theory is sound and the model is well validated with experiments. I therefore recommend this paper as a spotlight at NeurIPS.